Resource

# Cytokine storm and vascular leakage in severe dengue: insights from single-cell RNA profiling

Mohamad Al Kadi[1] , Maika Yamashita[1], Masayuki Shimojima[2], Tomoki Yoshikawa[2], Hideki Ebihara[2], Daisuke Okuzaki[1] , Takeshi Kurosu[2]

Severe dengue is characterized by vascular leakage triggered by a hyperinflammatory response, though the underlying mechanisms remain unclear. Our previous mouse model study highlighted the importance of small intestine in severe disease and identified key cytokines (IL-17A, TNF-$\alpha$, and IL-6) involved. Here, we used a Fixed RNA Profiling assay to characterize key cytokine- and effector-producing cells, along with their receptor expression. Type 3 innate lymphoid cells (ILC3), Th17 cells, and $\gamma\delta$ T cells emerged as pathologically relevant IL-17A/F-producing cells. These cells expressed IL-1$\beta$ and IL-23 receptors, underscoring the significance of these signaling pathways. IL-1$\beta$ was produced by M2-like macrophages, dendritic cells, and neutrophils, whereas M1-like macrophages, which differentiated post-infection, produced IL-23, TNF-$\alpha$, and IL-6, acting as initiators and amplifiers of the cytokine storm. Newly differentiated neutrophils produced IL-1$\beta$ and effector molecule matrix metalloprotease-8, suggesting a dual role in exacerbating the cytokine storm and directly mediating vascular leakage. Identified macrophages and neutrophils exhibited atypical characteristics. These findings provide new pathological insights into severe dengue and broader mechanism underlying cytokine storm-related diseases.

## Introduction

Dengue virus (DENV) infection is one of the most notable mosquito-borne diseases worldwide. An estimated 390 million dengue infections occur annually, of which ~96 million show clinical symptoms (Bhatt et al, 2013). The severity of dengue is often caused by vascular leakage resulting from cytokine storms, which can lead to organ failure (Wilder-Smith et al, 2019). Despite extensive research, the detailed pathogenic mechanisms involved in severe dengue remain unknown. Cytokine storms are a hallmark of severe infectious diseases (Cron et al, 2023) and pose significant treatment challenges and life-threatening risks. These storms can be broadly divided into two steps: cytokine- and effector-level events. In, cytokine-level events, infected cells produce cytokines that act as master regulators, whereas effector-level events in severe dengue involve cellular activation in response to elevated cytokine levels, increased vascular permeability in the periphery, and ultimately, organ damage.

To investigate host responses to pathogens and elucidate pathogenic mechanisms, bulk RNA sequencing (RNA-seq) and microarray analysis, which we performed previously (Kurosu et al, 2023), are invaluable tools. However, these methods assess gene expression at the whole-tissue or cell-population level, averaging the behavior of all cells and potentially masking the underlying heterogeneity. This limitation hinders a comprehensive understanding of the host response. Single-cell RNA-seq can overcome this challenge by detecting gene expression at the individual cell level. This powerful technology has rapidly advanced over the last decade, offering unprecedented insights into viral infections and host responses (Dance, 2022). Single-cell RNA-seq has revolutionized our ability to dissect pathogenic mechanisms with exceptional depth and speed (Zhao et al, 2021). Standard single-cell sequencing requires handling live cells, which induces cellular stress during preparation and demands complex experimental designs (Chen et al, 2018). These challenges are particularly pronounced in complex tissues such as organs, where extensive cell digestion and target cell enrichment may introduce potential bias and lead to RNA degradation. In cases of severe acute infections, organs damage and severe cellular stress typically lead to apoptosis (Kurosu et al, 2024). Recognizing these limitations, 10X Genomics has recently developed the Fixed RNA Profiling assay, a promising technology that aims to preserve RNA integrity in sensitive tissues by using fixation and hybridizing DNA probes to mRNA within the fixed cells, enabling gene expression analysis even when the original RNA has degraded.

We previously developed a mouse model of severe dengue, which exhibits pronounced vascular leakage driven by cytokine storms (Phanthanawiboon et al, 2016). A comprehensive and chronological analysis of this model identified the small intestine as a key organ for producing inflammatory cytokines and mediators (Kurosu et al, 2023). However, the specific cells responsible for producing cytokines and mediators remain unidentified. To address this, we employed a Fixed

[1]Laboratory of Human Immunology (Single Cell Genomics), WPI Immunology Frontier Research Center, Osaka University, Osaka, Japan   [2]Department of Virology I, National Institute of Infectious Diseases, Tokyo, Japan

Correspondence: dokuzaki@biken.osaka-u.ac.jp; kurosu@niid.go.jp

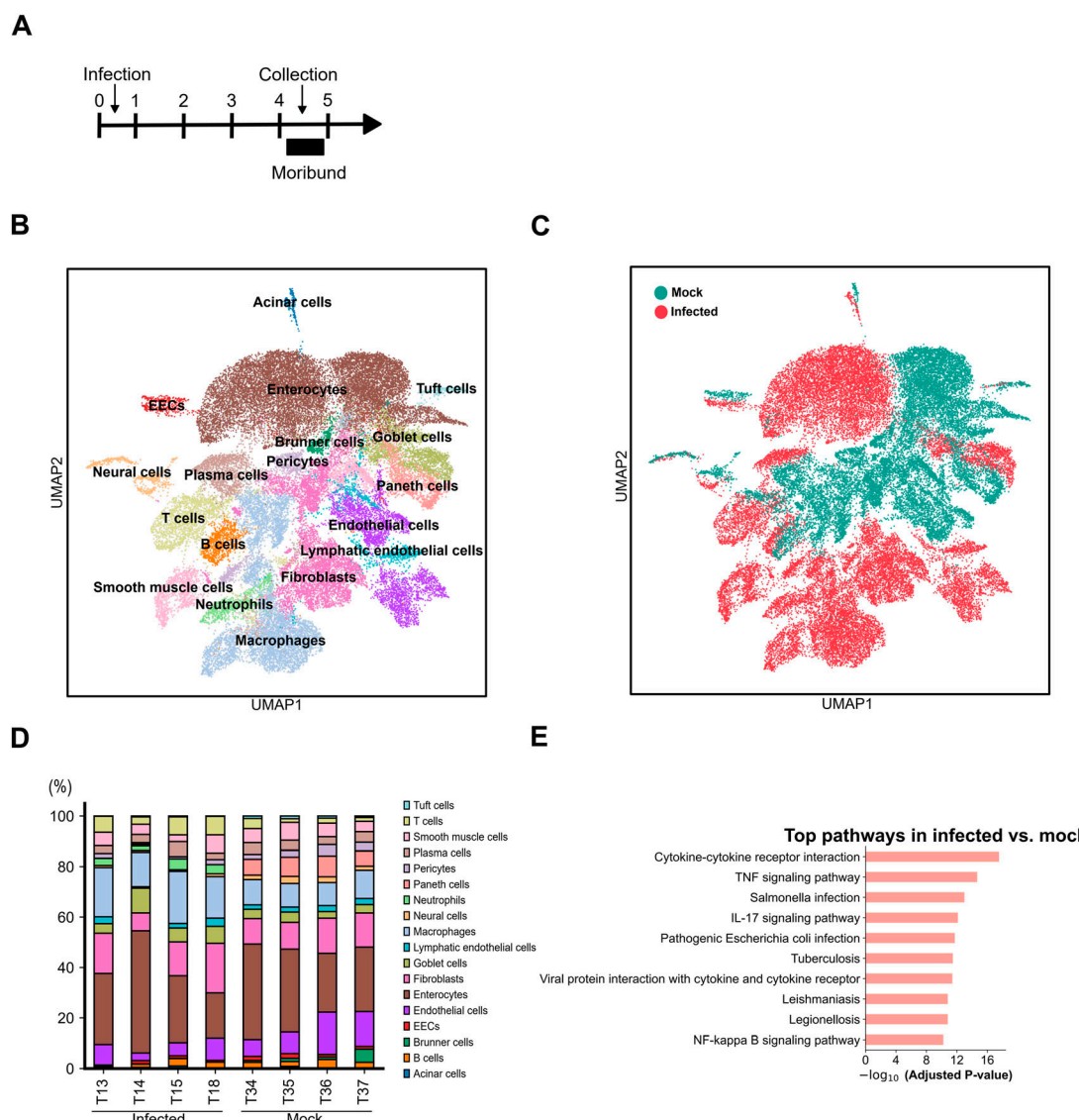

**Figure 1. Annotation and cellular composition of major intestinal cells following infection and mock infection.**
**(A)** IFN-$\alpha/\beta/\gamma$R KO mice (8–10 wk old) were intraperitoneally infected with $2 \times 10^6$ focus-forming units of DENV-3 P12/08 and euthanized at day 4 post-infection (p.i.). The small intestines were collected from infected (T13, T14, T15, and T18) and mock-infected mice (T34, T35, T36, and T37) for analysis. **(B)** Uniform Manifold Approximation and Projection plot of the four DV3P12/08-infected mice ($n = 4$) and mock-infected control ($n = 4$), which are colored by cell type. **(C)** Uniform Manifold Approximation and Projection plot showing integrated data, with cells colored according to sample origin (red for infected and green for mock-infected-mice). **(D)** Bar plot showing relative abundance of major cell populations in each sample. EEC, endocrine cell. **(E)** Top 10 enriched pathways identified by the KEGG pathway enrichment analysis of differentially expressed genes between infected and uninfected samples.

RNA Profiling assay to analyze RNA expression in the small intestine during severe dengue.

# Results

## Characterization of cell populations in the small intestines of DV3P12/08-infected mice

We characterized the overall dynamics of cell subpopulations from four DV3P12/08-infected mice at the moribund stage (Fig 1A)

compared with four mock-infected control using the Fixed RNA Profiling assay (also called the Fixed RNA Profiling assay by 10X Genomics [Pleasanton, CA, USA]). We obtained a total of 42,515 cells from the eight samples after quality control (Table S1). The data were normalized and scaled, and 3,047 highly variable genes were selected for further analysis. Cells were classified into 39 clusters based on their gene expression profiles using the Leiden algorithm, and these clusters were visualized in a 2D graph using Uniform Manifold Approximation and Projection (UMAP). Differential gene expression was analyzed, and the top genes in each cluster, along with canonical gene marker expression, were used to identify cell types (Figs 1B and S1).

The infection altered the composition of cell population composition (Fig 1C and D). Populations of immune cells such as T cells and macrophages were increased, whereas those of parenchymal cells were decreased by infection. Notably, the neutrophil population was drastically increased in all four mice during infection, whereas no neutrophils were detected in mock-infected mice (Fig 1D), which is consistent with our previous report (Kurosu et al, 2023). In addition, Paneth cells were lost during infection. Enterocytes constituted the largest population in infected mice, followed by macrophages (Fig 1D). To compare the gene expression profiles of infected and mock-infected mice, we performed pathway enrichment analysis based on the Kyoto Encyclopedia of Genes and Genomes gene sets. The cytokine-cytokine receptor interaction, TNF signaling pathway, and IL-17 signaling pathway were among the top 10 pathways (Fig 1E), highlighting the critical roles of cytokines, especially TNF-$\alpha$ and IL-17 signaling, in disease progression. These observations are consistent with our previous findings (Kurosu et al, 2023).

## Identification of IL-17A- and IL-17F-producing cells

For further investigation, we identified IL-17-producing cells. *Il17a/f* was highly produced in infected mice but not in mock-infected mice (Fig 2A). Owing to the limited number of *Il17a/f*-expressing cells in mock-infected mice, we focused on infected mice (Fig S2). Among all cell types, *Il17a* and *Il17f* expression was detected in T-cell population (Fig 2B and C). Sub-clustering and classifying T cells based on the expression of major genes revealed the presence of innate lymphoid cells (ILCs) (Figs 2D and S3). Type 3 innate lymphoid cells (ILC3) were categorized into three groups: ILC3, *Ifn$\gamma$*$^+$ILC3, and *Il17f*$^+$ILC3 (Fig S3). ILC3, which does not produce *Il17* nor *Ifn$\gamma$*, was detected only in mock-infected mice. *Ifn$\gamma$*$^+$ILC3 and *Il17f*$^+$ILC3 likely corresponded to *Ccr6*$^-$*Ncr*$^+$ILC3 and *Ccr6*$^+$*Ncr*$^-$ILC3, judging from their gene expression patterns (Fig S3) (Li et al, 2023). Infection increased the populations of ILC2, *Il17f*$^+$ILC3, *Cd8*$^+$ T cells, and regulatory T cells (Tregs) (Fig 2E). *Il17a* expression was identified in Th17 and $\gamma\delta$ T cells, whereas *Il17f* expression was observed in Th17, ILC3, and $\gamma\delta$ T cells (Fig 2F and G). Although IL-17A production in $\gamma\delta$ T cells had been previously reported (Kurosu et al, 2023), single-cell analysis newly identified Th17 cells and ILC3s as major IL-17A/F producers in this mouse model. Next, we investigated which signaling pathway possibly triggered these cells to produce IL-17A/F. TGF-$\beta$, IL-1$\beta$, IL-6, and IL-23 signaling are known to induce differentiation and activation of Th17 cells, $\gamma\delta$ T cells, and ILC3s (Korn et al, 2009; Sutton et al, 2009; Li et al, 2023). Therefore, we investigated the expression of receptors for these ligands on T cells. TGF-$\beta$ signaling is majorly mediated by TGF-$\beta$ type 1 (TGF$\beta$R1) and TGF$\beta$R2 receptors (Liu et al, 2018; Vander et al, 2018). *Tgf$\beta$r1* or *Tgf$\beta$r2* was expressed in almost all T cell types and ILCs, except for *Ifn$\gamma$*$^+$ILC3 (Fig 2H). IL-6 receptor $\alpha$ (*Il6ra*) was expressed in naïve T cells, but only weak expression was detected in IL-17A/F producers. Receptors of IL-1$\beta$ are IL-1R1 and IL-1R2 (Arend et al, 2008), and IL-1R2 is a decoy IL-1 receptor lacking the intracellular Toll/IL-1 receptor domain (Schluter et al, 2018). *Il1ra* and IL-23 receptor (*Il23r*) were expressed in all IL-17A/F-producers, including *Il17f*$^+$ILC3, Th17 cells, and $\gamma\delta$ T cells (Fig 2H). These findings suggested that TGF-$\beta$, IL-1$\beta$, and IL-23 signals likely drive the activation of IL-17A/F-producing cells, whereas IL-6 does not.

## CCR6 and CCL20 axis for Th17 and $\gamma\delta$ T cells

We next investigated the recruitment of *Il17f*$^+$ILC3s, Th17 cells, and $\gamma\delta$ T cells to the small intestine. These cells are originally abundant in the intestinal tract (Esplugues et al, 2011; Klose & Artis, 2020) but their numbers were substantially increased by infection in the small intestine of this model (Fig 2A). Besides, IL-17A-producing V$\gamma$4 and V$\gamma$6 TCR $\gamma\delta$ T cells, which expanded during infection, may have been recruited from other organs (Kurosu et al, 2023). The CCR6-CCL20 axis is important for cell migration to the intestine (Meitei et al, 2021). Although the proportion of *Ccr6*-expressing cells did not clearly increase in infected mice, their absolute numbers were elevated because of the overall expansion of IL-17-producing cells (Fig 3A). These recruited T cells, already expressing *Ccr6*, likely respond to CCL20 gradients to migrate to the site of infection. *Ccr6* expression was observed in T and B cells of infected mice (Fig 3B and C). Although the sub-clustering of T cell populations detected *Ccr6* expression in *Il17f*$^+$ILC3, ILC2, $\gamma\delta$ T, Th17, other Th cells and Treg cells, its expression was not prominent in *Il17f*$^+$ILC3 and $\gamma\delta$ T cells (Fig 3D and E). In contrast, *Ccl20* mRNA expression was significantly increased during infection (Fig 3F), and Brunner cells, enterocytes, goblet cells, and tuft cells were the major producers of CCL20 (Fig 3G and H). These results suggested that IL-17 producers likely migrated from other organs and the CCR6-CCL20 axis mediates their recruitment to the small intestine.

## Characterization of IL-6-expressing cells

A major effect of IL-17A was to enhance IL-6 production (Murakami & Hirano, 2011; Kurosu et al, 2023), which links the cytokine-level event to the effector-level event, such as vascular leakage and subsequent organ failure. We investigated IL-6-expressing cells and confirmed that *Il6* expression was increased in all four infected mice (Fig 4A). Single-cell analysis revealed that endothelial cells, fibroblasts, macrophages, and pericytes were the major IL-6 producers (Fig 4B and C). We then examined how IL-6 production was induced in these cells. *Il6* transcription is typically driven by the NF-$\kappa$B signaling pathway, which is activated by several pro-inflammatory cytokines (Dorrington & Fraser, 2019; Li et al, 2019). We previously reported that the infection induced TNF-$\alpha$, IL-17A, and IL-1$\beta$ expression, which potentially activates the NF-$\kappa$B signaling pathway (Kurosu et al, 2023). Therefore, we examined the expression of their receptors. All major IL-6-producers (endothelial cells, fibroblasts, macrophages, and pericytes) expressed one or both TNF-$\alpha$ receptors (*Tnfrsf1a* and *Tnfrsf1b*) (Fig 4D). In addition, *Il17ra* expression was detected in all cells except endothelial cells, albeit at different expression levels. *Il1r1* expression was observed in endothelial cells, fibroblasts, and pericytes but absent in macrophages. Conversely, *Il1r2* was expressed only in macrophages. Macrophages likely had different activation dynamics compared with other IL-6-producing cells, showing higher *Il17ra* expression but lacking *Il1r1* expression (Fig 4D). Interestingly, *Il6ra* was detected in fibroblasts, macrophages, and pericytes, suggesting a possible positive feedback effect. Collectively, TNF-$\alpha$ receptor was commonly expressed in these cells, with each cell type also expressing *Il17ra*, *Il1r1*, or *Il6ra*.

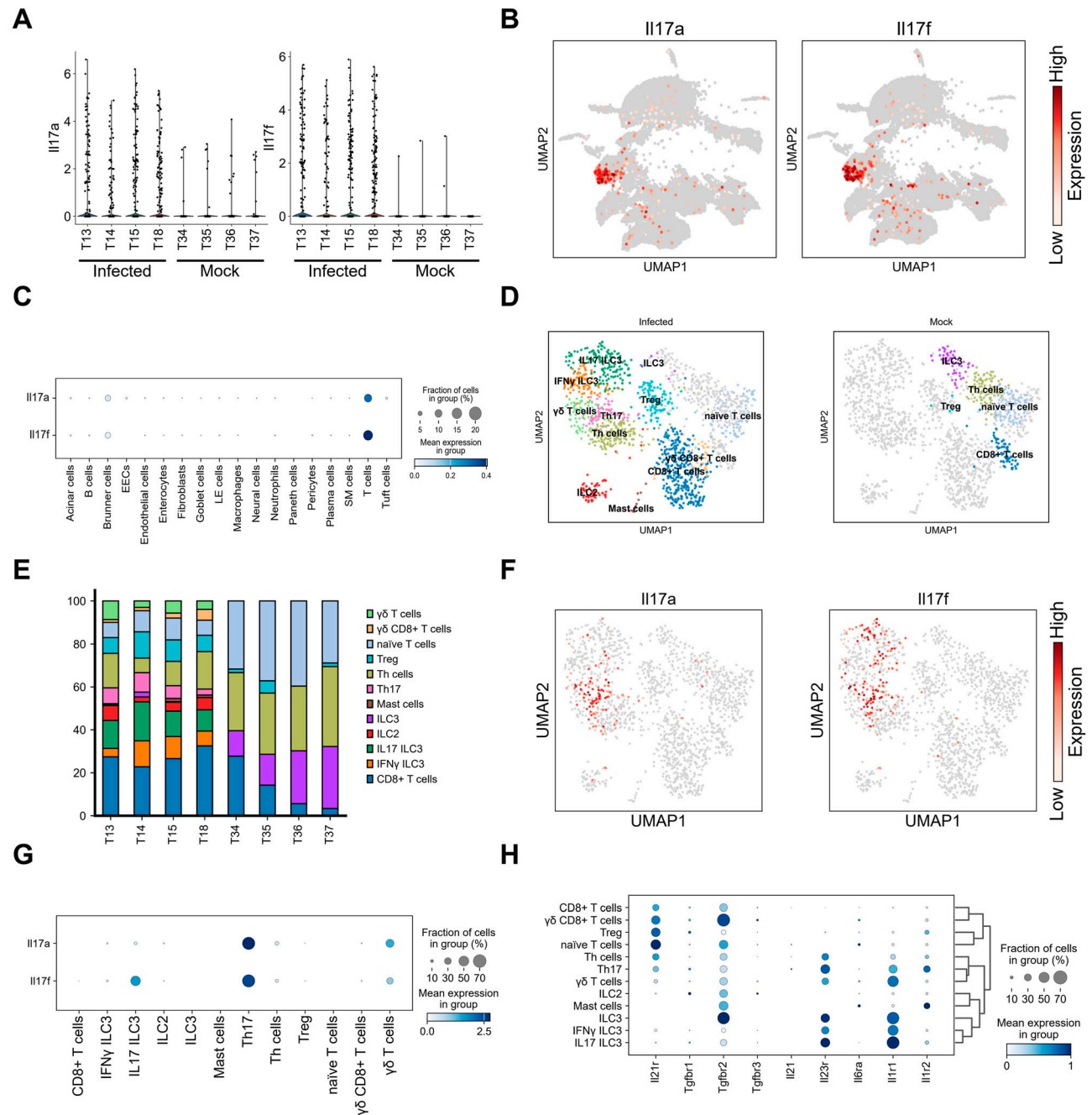

**Figure 2. *Il17a* is expressed in Th17 and γδ T cells.**
**(A)** Violin plot showing *Il17a* (left) or *Il17f* (right) expression levels of individual mouse infected with DV3P12/08 (T13, T14, T15, and T18) or mock-infected (T34, T35, T36, T37). **(B)** Uniform Manifold Approximation and Projection plot showing cell distribution of IFN-α/β/γR KO mice infected with DV3P12/08 (n = 4), colored *Il17a* (left) or *Il17f* (right) expression levels; cells with no expression are noncolored (gray), and those with measurable expression levels are colored on a scale from white (low) to red (high). **(C)** Dot plot showing *Il17a* and *Il17f* expression across intestinal cell types (n = 4). **(D)** Sub-clusters of intestinal T cells, including innate lymphoid cells (ILCs) from IFN-α/β/γR KO mice infected with DV3P12/08 (n = 4) (left) or mock-infected (n = 4) (right). **(E)** Relative abundance of T cell and ILC populations in each sample. **(F)** Uniform Manifold Approximation and Projection plot showing *Il-17a* (left) and *Il17f* (right) expression in T cells (n = 8). **(G)** Dot plot showing *Il17a* and *Il17f* expression in T cell and ILCs sub-clusters (n = 4). **(H)** Dot plot representing the expression of cytokines and their receptor genes, with dot color intensity indicating scaled mean expression. Scaling was relative to the expression of each gene for all cells in each annotation selection, that is, cells associated with each column label in the dot plot (n = 4).

Furthermore, we examined the coexpression of IL-6 and cytokine receptors in each IL-6 producer. Coexpression of TNF-α receptors (*Tnfrsf1a* and *Tnfrsf1b*) with *Il6* was widely detected in all *Il6*-producing cells (>30%) (Table 1). Over 30% coexpression of *Il1r1* with *Il6* was observed in endothelial cells, fibroblasts, and pericytes (51.0%, 57.8%, and 30.4%, respectively) but not in macrophages.

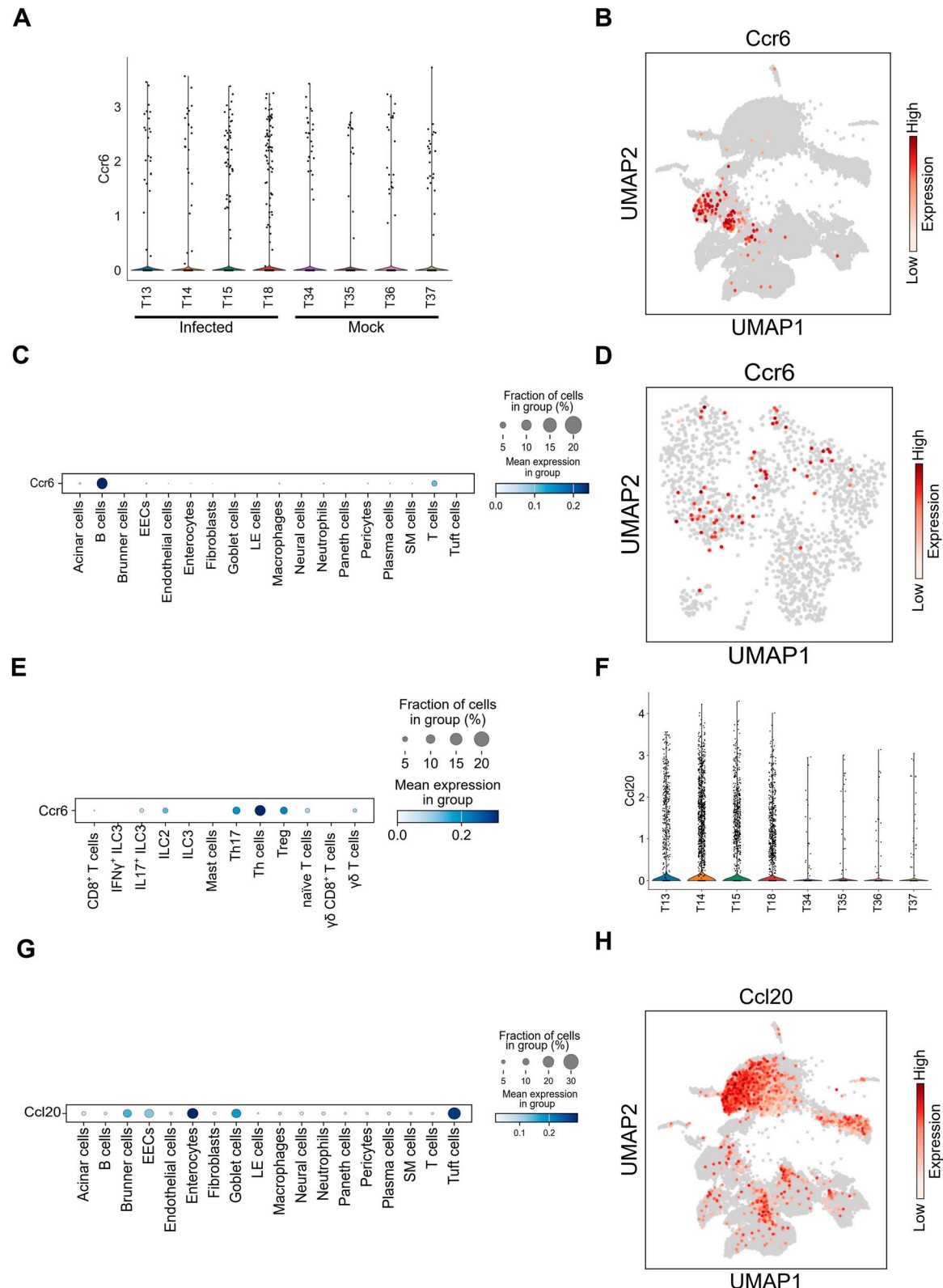

**Figure 3. CCR6-CCL20 axis.**
**(A)** Violin plot showing *Ccr6* expression scores for individual mouse infected with DV3P12/08 (T13, T14, T15, and T18) or mock-infected (T34, T35, T36, T37). **(B)** Uniform Manifold Approximation and Projection (UMAP) plot showing cell distribution of IFN-α/β/γR KO mice infected with DV3P12/08 (*n* = 4), colored *Ccr6* expression level; cells with no expression are noncolored (gray), and those with measurable expression levels are colored on a scale from white (low) to red (high). **(C)** Dot plot showing the

Expression of *Il17ra* was detected in all cell types, with the highest in macrophages (25.1%). *Tnfrsf1a* or *Tnfrsf1b* and *Il1r1* were highly expressed in all groups, but the expression patterns of other receptors were different. Based on receptor expression patterns, IL-6-producing cells could be divided into two groups: the first group, including endothelial cells, fibroblasts, and pericytes, exhibited moderate *Il17ra* expression; the second group, a small proportion of macrophages, expressing *Il1r1* (6.8%). A relatively higher percentage of macrophages expressed *Il17ra*. Notably, TNF-α signaling appeared critical for both groups, with IL-1 signaling enhancing the activation of the first group and IL-17 signaling predominantly affecting macrophages.

### IL-1β production from macrophages and neutrophils

The above results indicated the strong involvement of IL-1β in both IL-17A/F and IL-6 production. *Il17*-producing ILC3s, Th17 cells, and γδ T cells highly expressed *Il1r1* (Fig 2H). Infection apparently increased the number of *Il1β*-producing cells (Fig 5A). Therefore, we screened for *Il1β*-expressing cells and identified dendritic cells (DCs), macrophages, and neutrophils as its major producers (Fig 5B and C). We also observed stronger inflammatory responses in macrophages and neutrophils (Fig S4), suggesting their significant involvement in infection-driven inflammation. To identify distinct macrophage subtypes along with DCs, we sub-clustered them using the Leiden algorithm followed by UMAP visualization. This yielded three macrophage clusters (M1-like macrophages, M2-like macrophages, and monocytes) and DCs (Fig 5D–F). M2-like macrophages were detected only in mock-infected mice, whereas infection induced M1-like macrophages and monocytes (Fig 5D and E). *Il1β* expression was detected in M2-like macrophages, monocytes, and DCs but to a lesser degree in M1-like macrophages (Fig 5G). In contrast, non-*Il1β* producer M1-like macrophages expressed *Tnfα* and *Il6* (Fig 5F). High levels of antiviral genes were detected in M1-like macrophages (Figs 5H and S5), suggesting their potential as viral infection target.

Sub-clustering neutrophils using the Leiden algorithm resulted in eight clusters (0–7) (Fig 6A). Of note, sub-clustering was performed exclusively on infected samples because no neutrophils were identified in mock-infected mice. Neutrophils are usually present as a single spectrum in single-cell RNA-seq data. This spectrum corresponds to the developmental stages of neutrophils (Xie et al, 2020). Although there is a current dispute regarding neutrophil classification via single-cell analysis (Xie et al, 2020), demonstrated the classification of clusters ranging from G0–G5c using uninfected mice. Clusters G0–G4 represent differentiated bone marrow cells, whereas G5a–G5c represent peripheral blood neutrophils. Based on this classification, we analyzed eight sub-clusters. These neutrophils expressed G4, G5a, and G5c genes (Figs 6B and S6A and B), suggesting that they have the characteristics of both immature bone marrow and mature peripheral blood neutrophils, unlike typical peripheral neutrophils. Two lineages

were identified: one group containing sub-clusters 1, 2, and 4, and the other containing sub-clusters 3, 5, 6, and 7 (Fig 6A). All neutrophil sub-clusters except sub-cluster 0 expressed *Il1β* (Fig 6C). Sub-cluster 0 contained the most immature neutrophils among the eight sub-clusters (Fig 6B) and primarily expressed G2 and G3 genes (Figs 6B and S6).

We further examined cytokine receptor expression in IL-1β-producing cells. In macrophages/DCs, all IL-1β-producers (M2-like macrophages, monocytes, and DCs) expressed *Tnfrsf1a* and *Tnfrsf1b* but IL-1β-nonproducers (M1-like macrophages) did not (Fig 5I). *Il17ra* was exclusively expressed in M1-like macrophages. All IL-1β producers expressed *Il1r2* (Fig 5I). In contrast, in neutrophils, *Il17ra*, *Tnfrsf1a*, and *Tnfrsf1b* were expressed in almost all sub-clusters (Fig 6D). In addition, neutrophils highly expressed *Cxcr1* and/or *Cxcr2*, which are crucial for neutrophil recruitment and activation (Capucetti et al, 2020).

IL-1β is produced as a pro-form and requires cleavage by caspase-1 for its activation (Schett et al, 2016). Using this mouse model, we had already reported high serum levels of IL-1β in infected mice (Kurosu et al, 2023), indicating significant production and release of mature IL-1β. This suggests the involvement of inflammasome in pro-IL-1β cleavage by caspase-1 and release (Schett et al, 2016). Activation of the inflammasome also induces transcription of NLR family pyrin domain-containing 3 (NLRP3), which is a major component of the inflammasome (Wang et al, 2024). In this model, high *Nlrp3* expression was exclusively detected in macrophages and neutrophils (Fig S7A and B), suggesting the possible involvement of inflammasome in IL-1β production.

### IL-23 production by M1-like macrophages

In addition to IL-1β, IL-23 is a key activator of IL-17A/F-producing *Il17f*⁺ILC3, γδ T cells, and Th17 cells as they expressed *Il23r* (Fig 2H). Macrophages exhibited the highest *Il23a* expression among all cell types (Fig 7A). Further investigation revealed that *Il23a* expression was exclusively detected in M1-like macrophages (Fig 7B). These results suggested that M1-like macrophages act as initiators to activate other cells, including IL-17A/F-producing cells, IL-1β-producing M2-like macrophages, monocytes, and DCs.

### Matrix metalloprotease (MMP)-8-expressing neutrophils

To examine effector-level events, we explored the role of MMP-8, as we had previously identified it as a candidate effector molecule that could induce vascular leakage (Kurosu et al, 2023). Given this finding, we hypothesized that neutrophils are the major players that induce vascular leakage and confirmed that *Mmp8* production was exclusive to neutrophils (Fig 8A). MMP-9 has also been implicated in vascular leakage (Luplertlop et al, 2006). Although neutrophils produce *Mmp9* during infection (Fig 8A), our previous

---

scaled expression of *Ccr6* gene across all cell clusters, colored by the average expression of *Ccr6* (*n* = 4). **(D)** UMAP plot of *Ccr6* gene expression in T cell and ILC sub-clusters (*n* = 4). **(E)** Dot plot showing *Ccr6* expression levels in T cell and ILCs sub-clusters of infected mice (*n* = 4). **(F)** Violin plot showing *Ccl20* expression levels for individual mouse infected with DV3P12/08 (T13, T14, T15, and T18) or mock-infected (T34, T35, T36, T37). **(G)** Dot plot showing *Ccl20* expression in cells derived from DV3P12/08-infected mice (*n* = 4). **(H)** UMAP plot representing *Ccl20* expression in small intestine cells of infected mice (*n* = 4).

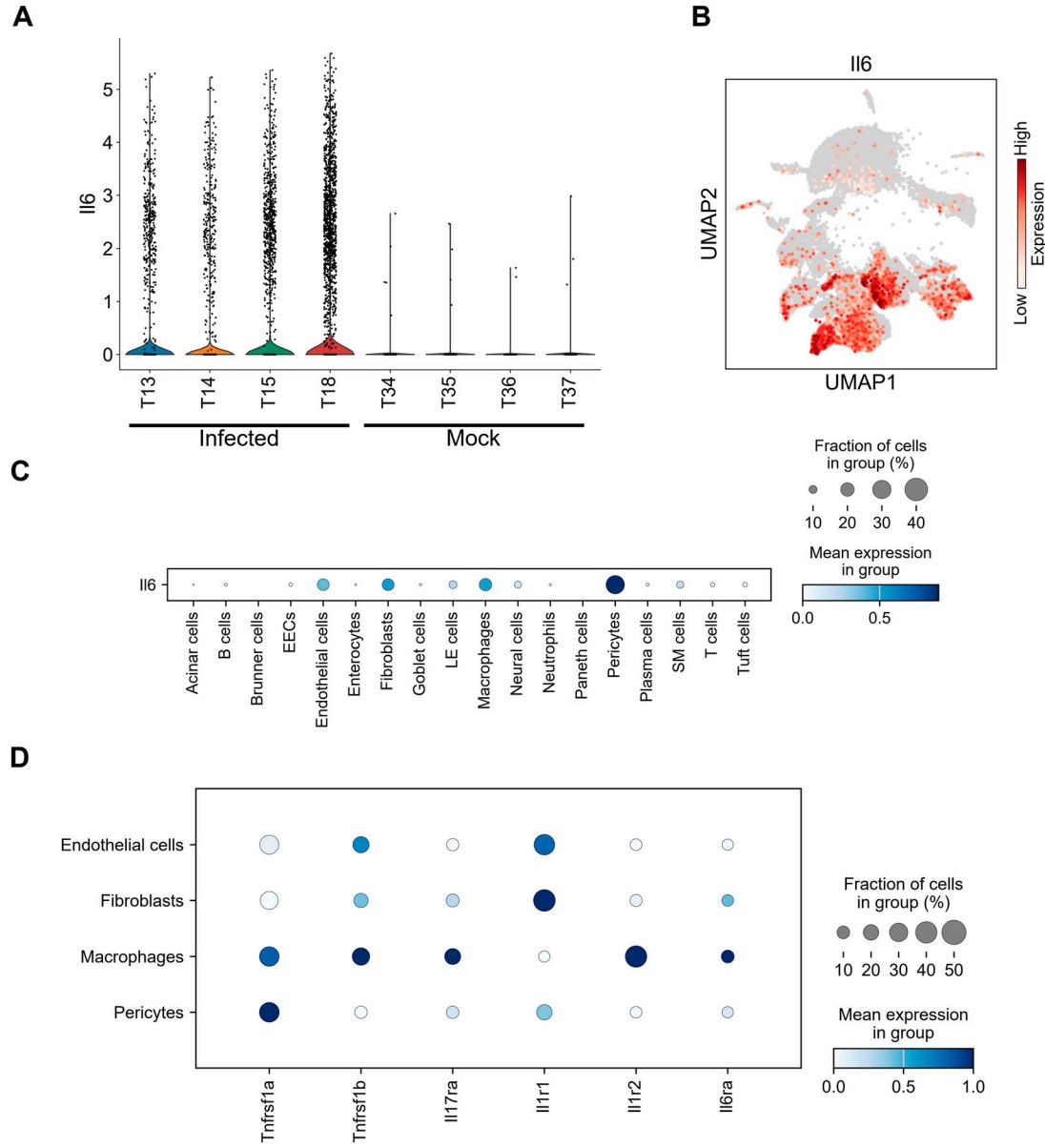

**Figure 4.** *Il6* **and receptor expression in IL-6-producing cells.**
**(A)** Violin plot showing I*Il6* expression scores for individual mice infected with DV3P12/08 (T13, T14, T15, and T18) or mock-infected (T34, T35, T36, T37). **(B)** Uniform Manifold Approximation and Projection plot showing integrated single-cell intestinal cell profiles from the four DV3P12/08-infected mice. *Il6* gene expression was color-coded from low to high expression (grey to red, respectively) (right) (*n* = 4). **(C)** Dot plot showing the scaled expression of *Il6* gene across all cell types, colored by the average expression of *Il6* (*n* = 4). **(D)** Dot plot showing the scaled expression of *Tnfrsf1a*, *Tnfrsf1b*, *Il17ra*, *Il1r1*, *Il1r2*, and *Il6ra* across cell types, with color intensity indicating expression levels (*n* = 4).

findings demonstrated that MMP-9 expression does not increase during infection (Kurosu et al, 2023), indicating its consistent expression by neutrophils under normal conditions. Expression levels of MMPs were standardized to all cells from infected mice because no neutrophils were detected in mock-infected mice. Among the identified neutrophil sub-clusters, sub-clusters 0, 3, 5, 6, and 7 expressed *Mmp8* (Fig 8B and C). These sub-clusters fell within the same single spectrum (Fig 6A). Receptor expression analysis of the major cytokines and chemokines revealed that neutrophils expressed *Tnfrsf1a*, *Tnfrsf1b*, *Il17ra*, *Il6ra*, and *Cxcr2* (Fig 8D).

Furthermore, analysis of *Mmp8*-producing neutrophils revealed coexpression with key receptors. *Tnfrsf1a*, *Tnfrsf1b*, and *Il17ra* were expressed in 30.1%, 23.1%, and 37.6% of these neutrophils, respectively (Table S2). In addition, 10.7% of *Mmp8*-expressing neutrophils expressed *Il6ra*. The inhibitory receptor, *Il1r2*, was highly expressed, suggesting a lack of IL-1β signaling. Furthermore, 41.5% of the *Mmp8*-producing neutrophils expressed *Cxcr2*. Notably, only a weak antiviral gene response was observed (Fig S8A and B). These results suggested that *Mmp8*-expressing neutrophils were activated by TNF-α, IL-6, IL-17A, and CXCR2 in this mouse model.

**Table 1.** Number of cytokine receptor-expressing cells in *Il6*-expressing cells.

| Name | *Il6*-expressing cells | *Tnfrsf1a* | *Tnfrsf1b* | *Il1r1* | *Il1r2* | *Il17ra* | *Il6ra* |
|---|---|---|---|---|---|---|---|
| Endothelial cells | 257 | 107 | 74 | 131 | 23 | 31 | 14 |
| % | | 41.6 | 28.8 | 51.0 | 8.9 | 12.1 | 5.4 |
| Fibroblast | 495 | 156 | 75 | 286 | 40 | 74 | 32 |
| % | | 31.5 | 15.2 | 57.8 | 8.1 | 14.9 | 6.5 |
| Pericytes | 102 | 39 | 11 | 31 | 9 | 8 | 5 |
| % | | 38.2 | 10.8 | 30.4 | 8.8 | 7.8 | 4.9 |
| Macrophages | 707 | 264 | 184 | 48 | 224 | 177 | 69 |
| % | | 37.3 | 26.0 | 6.8 | 31.7 | 25.1 | 9.8 |

# Discussion

This study is the first to investigate small intestine at the single-cell level during DENV infection, revealing detailed characteristics of each cell. Our analysis focused on key cytokines, including TNF-*α*, IL-6, IL-1*β*, IL-17, and IL-23, which have been frequently reported to be elevated in infected individuals (Bozza et al, 2008; Becquart et al, 2010; Guerrero et al, 2013; Sanchez-Vargas et al, 2020; Puc et al, 2021). Mouse model studies have also shown that DENV infection induces *γδ* T cells and NK cells to produce IL-17A in the spleen (Guabiraba et al, 2013). In our previous study using mouse model, we observed that large numbers of *γδ* T cells infiltrated the intestinal tract and produced IL-17A (Kurosu et al, 2023). However, there was no consensus on this phenomenon. In this study, using single-cell analysis, we identified Th17 cells as IL-17A producers as well as *γδ* T cells. ILC3s and Th17 cells were also identified as exclusive IL-17F producers (Fig 2E and F). This successful identification of Th17 cells and ILC3, which was not possible in previous studies, can be attributed to procedural improvements. Specifically, immediate freezing of the small intestine in liquid nitrogen after collection, followed by rapid fixation before thawing helped prevent cell death and allowed cellular detection under high infection-induced stress. Although IL-17A and IL-17F have protective roles against certain infections, they are also key pathogenic cytokines in T cell-mediated autoimmune disease pathology (Mills, 2023). In our mouse model, IL-17A exacerbated disease (Kurosu et al, 2023). Both Th17 and *γδ* T cells produced *Il17a* and *Il17f*, whereas ILC3 exclusively produced *Il17f*. IL-17A and IL-17F can form homodimers or heterodimers, and IL-17 signaling is regulated at the receptor level, which leads to differential effects across various organs (Tout et al, 2023). While IL-17A has been extensively studied, the role of IL-17F remains less clear (Cole et al, 2023). Furthermore, investigations are needed to determine whether IL-17A or IL-17F have distinct functions in IL-17 signaling.

Innate IL-17 producers, such as ILC3 or *γδ* T cells, reside in the small intestine, and require 4–8 h to produce IL-17 after stimulation with pathogen-associated molecular patterns or other stimulants in the presence of IL-1*β* and IL-23 (Cua & Tato, 2010; Omenetti et al, 2019; Li et al, 2023). In contrast, Th17 cells differentiate from CD4+ naïve T cells over require 3–5 days in the presence of TGF-*β*, IL-6, and IL-1*β* (Margelidon-Cozzolino et al, 2022). IL-23 is necessary to stabilize Th17 cells to produce IL-17. Although *Tgfβrs*, *Il6ra*, and *Il1r1* were expressed on Th17 cells (Fig 2H), it is likely that some of the IL-

17-producing Th17 cells detected in this study were already differentiated and residing in the small intestine before infection. There are likely two types of Th17 cells in the intestine: tissue-resident homeostatic Th17 cells that differentiate in response to the microbiota, and inflammatory Th17 cells (Omenetti et al, 2019). Both cell populations can be activated by IL-1*β* and IL-23 to produce IL-17A/F (Mills, 2023). Our study confirmed that all IL-17-A/F producers highly expressed *Il1r1* and *Il23r* (Fig 2H). Thus, IL-1*β* and IL-23 are critical for inducing Il-17A/F production.

An important finding of this study is that not only TNF and IL-17 but also IL-1*β* play important roles in disease progression, as indicated by receptor expression in *Il17a*-producing cells (Fig 2H) and *Il6*-producing cells (Fig 4D). The major *Il1β*-producing cells identified were macrophages and neutrophils (Fig 5B and C). *Il1β* production was unlikely triggered by direct viral infection (Figs 5I and S5), as neutrophils were not infected (Kurosu et al, 2023). This suggests that IL-1*β* production is a secondary response after the initial infection event. M1-like macrophages are generally thought to respond to infections at an early stage by producing cytokines. Prolonged or excessive M1-like macrophage activation is often linked to cytokine storms (Yu et al, 2022). Conversely, M2-like macrophage are believed to prevent inflammation-induced tissue damage during the late stages of disease by suppressing or ceasing inflammation. Therefore, cytokine storm could be attributed to either failure or delay in the emergence of M2-like macrophages. IL-1*β*, a major inflammatory cytokine, is thought to be produced by M1-like macrophages (Wang et al, 2024). However, in this model, M2-like macrophages were the center of IL-1*β* production (Fig 5H), suggesting that these cells may amplify inflammation rather than suppress it. These unique M2-like macrophages need to be further characterized, as they may play an important role in cytokine storms.

In this model, TNF-*α* signaling was the most important upstream master regulator, because its blockade completely protected mice from lethal infection and suppressed the induction of most cytokines such as IL-1*β*, IL-17A, IL-6, and IL-12p70 (Kurosu et al, 2023). IL-1*β* production appeared to be triggered by TNF-*α*, produced by M1-like macrophages (Fig 5G). Another factor, *Il23a*, was expressed by M1-like macrophages (Fig 7B), which corresponds to the characteristics of M1 macrophages (Wang & Karin, 2015). This is also consistent with the fact that blocking TNF-*α* signaling does not inhibit IL-23 production in this model (Kurosu et al, 2023). These observations with TNF-*α* and IL-23 indicate that M1 macrophages

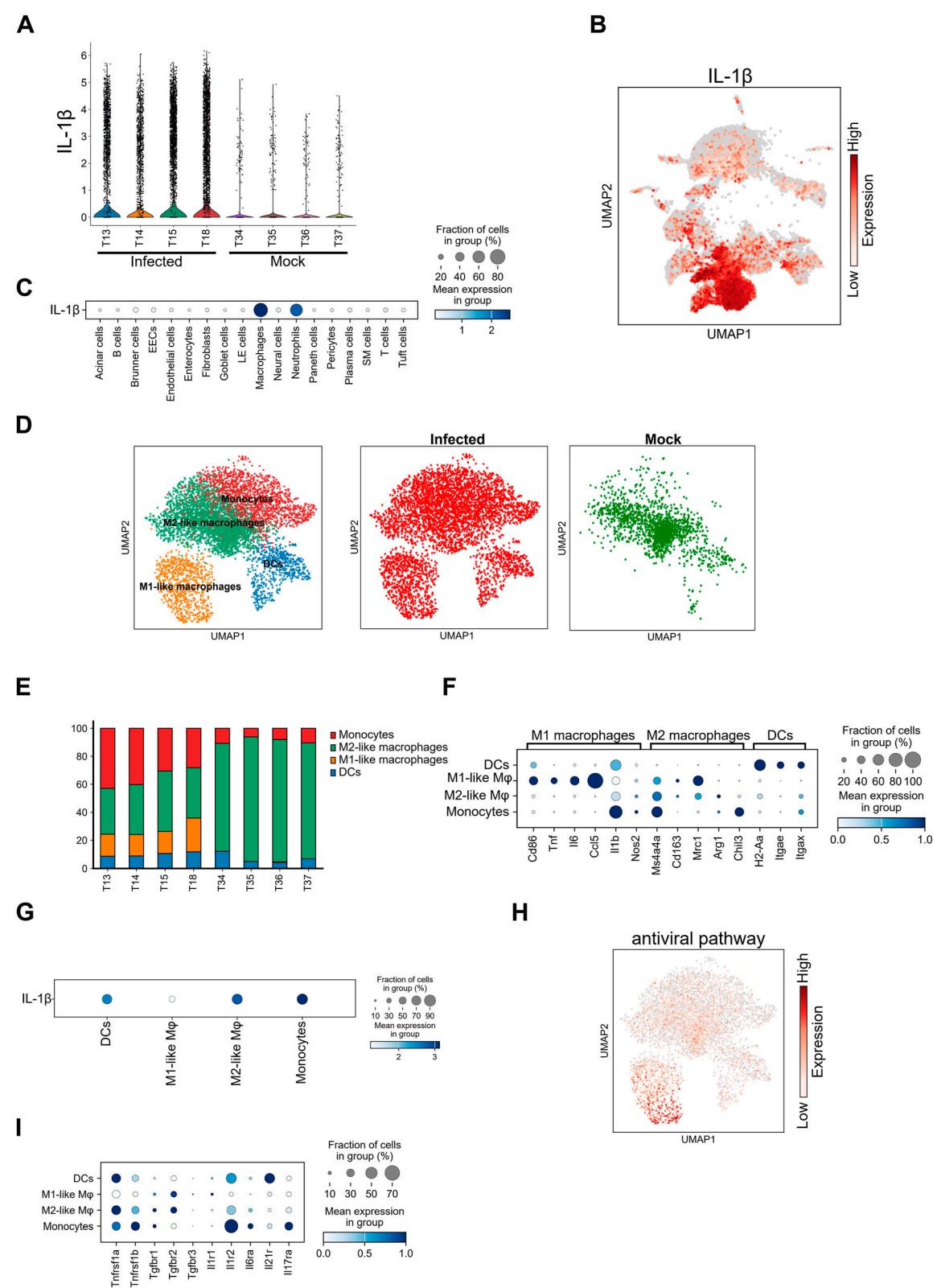

**Figure 5. _Il1β_ expression in macrophages and neutrophils.**

**(A)** Violin plot showing _Il1β_ expression scores for individual mouse infected with DV3P12/08 (T13, T14, T15, and T18) or mock-infected (T34, T35, T36, T37). **(B)** Uniform Manifold Approximation and Projection (UMAP) plot showing integrated single-cell intestinal cell profiles from the four DV3P12/08-infected mice. _Il1β_ gene expression was color-coded from low to high expression (grey to red, respectively) (right) (_n_ = 4). **(C)** Dot plot showing the scaled expression of _Il1β_ in all cell types, colored by the

are central initiators of cytokine storms by directing IL-1β and IL-17A/F production. Macrophages are diverse and plastic, which makes it challenging to classify them strictly as M1 and M2. M1-like- and M2-like macrophages identified in this study may represent unique infection-induced subtypes that are key to understanding the mechanism of cytokine storm. Future studies should focus on elucidating the differentiation pathways and inflammatory roles of these macrophages. Collectively, M1-like macrophages are likely to be initially activated to produce TNF-α and IL-23, followed by IL-1β production by M2-like macrophages, DCs, and neutrophils.

One of major roles of IL-17A is to enhance IL-6 production through synergy with TNF-α (Murakami & Hirano, 2011). IL-6 is the key mediator that connects cytokine-level events to effector-level events (Tanaka et al, 2016; Kurosu et al, 2023). However, based on receptor expression patterns, IL-17A/F may not be essential for this process. IL-6-expressing cells can be classified into two groups, both of which commonly express *Tnfrsf1a* or *Tnfrsf1b*. In addition, endothelial cells, pericytes, and fibroblasts expressed *Il1r1* (Fig 4D, Table 1), whereas macrophages highly expressed *Il17ra* but not *Il1r1*. This suggests that IL-17 or IL-1 may synergistically function with TNF-α in various cell types, thereby maximizing IL-6 production. Importantly, TNF-α was confirmed to be a major contributor to IL-6 production. A major function of IL-6 is presumed to be the activation of neutrophils. Indeed, both *Il6r* and *Il17ra* were sufficiently expressed on neutrophils (Fig 6D). IL-17 signaling plays a significant role in neutrophil activation (Iwakura & Ishigame, 2006). Moreover, *Il17ra* expression was also observed in neutrophils, B cells, and plasma cells (Fig S9A). IL-17 is known to induce B-cell proliferation and differentiation into IgG-secreting plasma cells (Zhu & Qian, 2012). Thus, IL-17 signaling may play an additional role in antibody production within this model, although the specifics remain unclear. The function of IL-17 signaling in B cells during DENV infection in both mice and humans warrants further elucidation.

Another important finding of this study is the role of neutrophils. Neutrophils have traditionally been considered effector or mediator cells involved in cytokine storm (Chan et al, 2021). They have also been reported to induce vascular leakage during dengue viral infection by forming neutrophil extracellular traps (Sung et al, 2019). In contrast, less attention has been given to their role as cytokine storm inducers despite their capacity to produce pro-inflammatory cytokines. This study strongly indicates that neutrophils are not merely effector or mediator cells causing tissue damage but also function as IL-1β producers. The infiltrated neutrophils showed complex features (Fig 6B), unlike those of typical G5a–G5c peripheral neutrophils. Premature neutrophils often appear in the periphery during infection or inflammation (Capucetti et al, 2020) and exhibit significantly

different cytokine and secretory protein gene expression patterns (Xie et al, 2020). Furthermore, neutrophils secreted several molecules that modulate inflammatory response, such as S100 calcium-binding protein A8 (S100A8) and S100A9 (Fig S6A). These proteins can potentially enhance cytokine production by stimulating Toll-like receptor 4 or the receptor for advanced glycation end-products (Wang et al, 2018). In this mouse model, TLR4 signaling pathway was strongly activated (Kurosu et al, 2023). These observations suggested that neutrophils may play an important role in cytokine storms by enhancing cytokine production. Although few studies have focused on neutrophils in DENV infection, neutrophilia has been reported in patients with severe dengue (Her et al, 2017). Further studies focusing on neutrophils are required to better understand the mechanisms underlying severe infectious diseases.

We proposed a model that illustrates cell-to-cell interactions inferred from receptor and ligand expression (Fig 9). In the initial step, the major event involves M1-like macrophages activation, which leads to TNF-α and IL-23 secretion. TNF-α stimulates M2-like macrophages/monocytes and DCs to produce IL-1β. IL-1β and IL-23 stimulate IL-17A/F-producing cells (ILC3s, γδ T cells, and Th17 cells). The produced IL-17A/F play several roles: (1) IL-17A/F, together with TNF-α, enhances the activation of endothelial cells, epithelial cells, fibroblasts, and pericytes, thereby inducing IL-6 production; (2) IL-17A/F recruits and activates neutrophils, inducing IL-1β and MMP-8 production; (3) IL-17A/F likely further activates M1- and M2-like macrophages/monocytes, creating a positive feedback loop; and (4) IL-17A/F may also affect B and plasma cells and presumably influence antibody production. IL-6 produced by pericytes, fibroblasts, endothelial cells, and macrophages enhances neutrophil activation. Neutrophils are further activated by several factors. The CSF3 receptor (CSF3R), which promotes neutrophil proliferation, was highly expressed in neutrophils (Fig S9B). The ligand CSF3 was expressed by endothelial cells, fibroblasts, and lymphatic endothelial cells. Endothelial cells, epithelial cells, fibroblasts, and pericytes also produced CSF3 (G-CSF) and CXCL1 (Fig S9C and D), which induced the proliferation, differentiation, recruitment, and activation of neutrophils through the CSF3R and CXCR2 expressed on neutrophils. Consequently, a large number of activated neutrophils functioned as effector cells to exacerbate disease in the small intestine.

As described above, at the single-cell level, a wide variety of immune cells are activated by infection, interact with non-immune cells, undergo differentiation and proliferation, and are actively recruited. Notably, these processes occur within the intestinal tract. Recently, the intestinal tract has gained attention for its potential involvement in systemic conditions driven by immune overreactions, such as systemic inflammatory response syndrome and cytokine storm syndrome (Chan et al, 2021; Murao et al, 2023). Given

---

average expression of *Il1β* (n = 4). **(D)** Sub-clusters of the intestinal macrophages (Mφ) from IFN-α/β/γR KO mice infected with DV3P12/08 (n = 4) and mock-infected (n = 4). UMAP plot showing integrated data and colored according to sample origin (red for infected and green for mock-infected). **(E)** Relative abundance of M1-like- and M2-like macrophages, monocytes, and DCs in each sample from the individual mouse infected with DV3P12/08 (T13, T14, T15, and T18) or mock-infected (T34, T35, T36, T37). **(F)** Dot plot showing the expression levels of gene markers used to annotate M1-like- and M2-like macrophages, monocytes, and DCs (n = 4). **(G)** Dot plot showing *Il1β* gene expression in M1-like- and M2-like macrophages, monocytes, and DCs (n = 4). **(H)** UMAP plot of antiviral gene expression in M1-like- and M2-like macrophages, monocytes, and DCs (n = 4). **(I)** Dot plot showing cytokine receptor gene expressions in M1-like- and M2-like macrophages, monocytes, and DCs (n = 4). Dot color represents scaled mean expression.

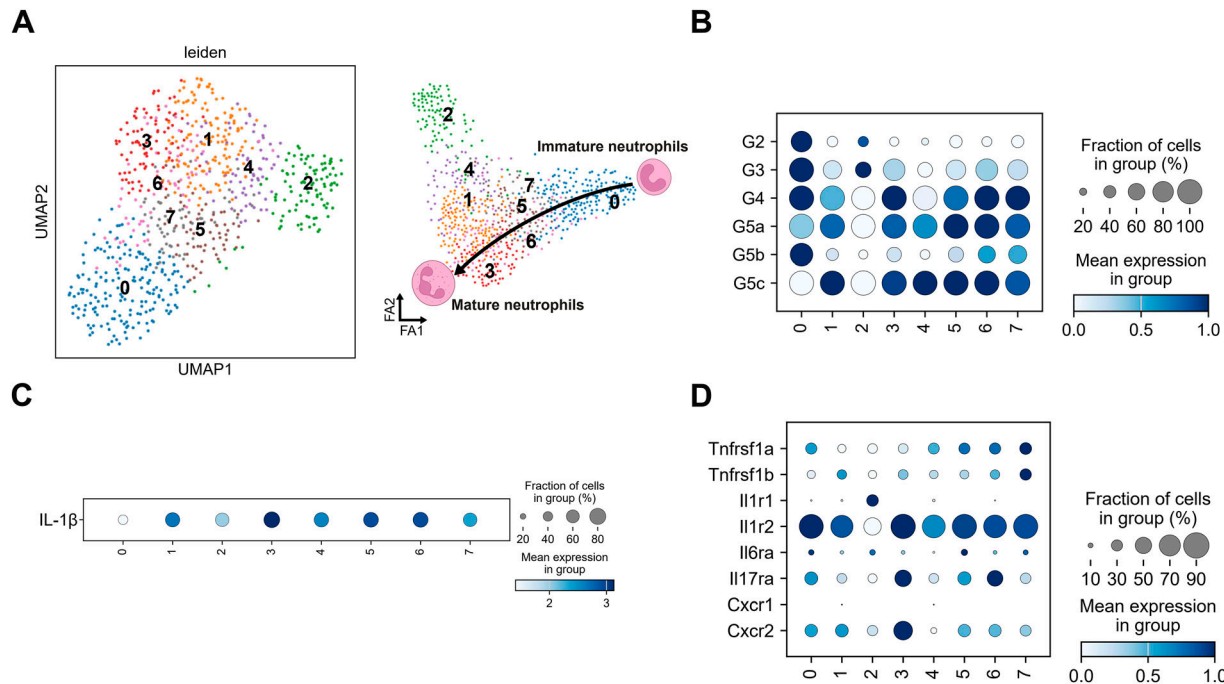

**Figure 6.** *Il1β* **expression in neutrophil sub-clusters.**
**(A)** Sub-clusters of intestinal neutrophils from IFN-α/β/γR KO mice infected with DV3P12/08 (*n* = 4) (left) and trajectory analysis of neutrophils (right). **(B)** Dot plot showing representative gene expression in neutrophil sub-clusters (*n* = 4). **(C)** Dot plot showing *Il1β* gene expression in neutrophil sub-clusters (*n* = 4). **(D)** Dot plot showing cytokine receptor gene expression in neutrophil sub-clusters (*n* = 4). Dot color represents scaled mean expression.

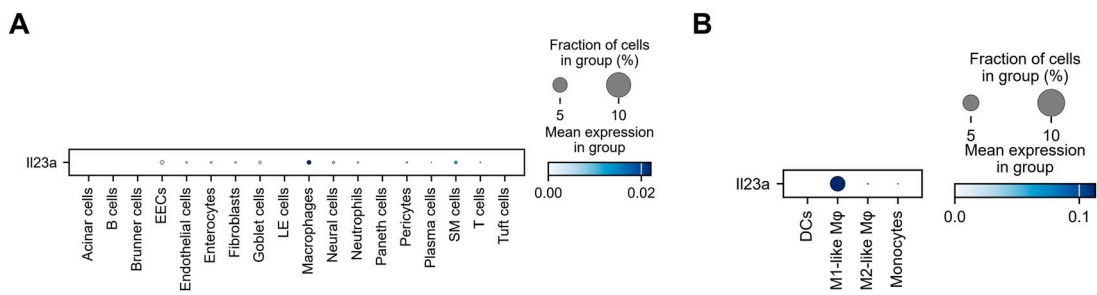

**Figure 7.** *Il23a* **expression in M1-like macrophages.**
**(A)** Dot plot showing scaled expression of *Il23a* gene, colored by the average *Il23a* expression in all cell types (*n* = 4). **(B)** Dot plot depicting *Il23a* gene expression in M1-like- and M2-like macrophages, monocytes, and DCs (*n* = 4). Dot color represents scaled mean expression.

that the small intestine represents the largest secondary lymphoid tissue, it may play a critical role in these phenomena. Further research from new perspectives is essential to deepen our understanding of these mechanisms.

Overall, this study provides valuable insights into infection-induced cell properties. However, this study has certain limitations. Firstly, our analysis was performed at a single time point, which may have missed earlier immune events. Secondly, we still do not know whether these observations are consistent with human dengue cases. Thirdly, the Fixed RNA Profiling assay is unable to distinguish between unsliced and spliced mRNA, thus preventing RNA velocity analysis difficult for deeper cell trajectory analysis (La Manno et al, 2018). Nevertheless, this study provides novel insights into the role of individual cell types. Future single-

cell-level studies could further elucidate the role of cytokines in dengue severity.

# Materials and Methods

## Ethics statement

All experiments involving animals were performed in animal biological safety level 2 containment laboratories at the National Institute of Infectious Diseases (NIID), Japan, in accordance with the animal experimentation guidelines of the NIID. All protocols were approved by the Institutional Animal Care and Use Committee of the NIID (nos. 121003 and 121005). Trained laboratory personnel

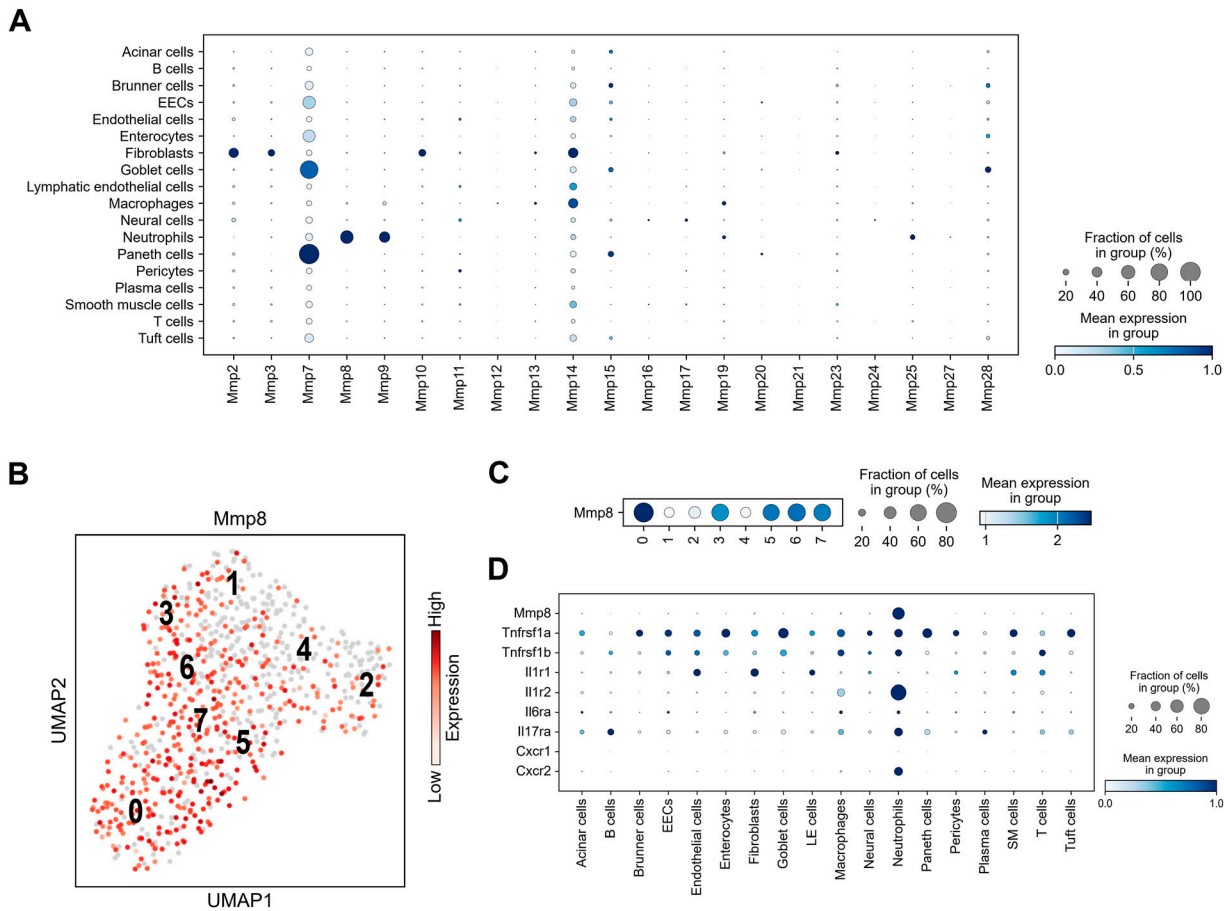

**Figure 8. Matrix metalloproteinase (*Mmp*)-8 expression in neutrophils.**
**(A)** Dot plot showing *Mmp* expression in cells derived from the small intestines of mice infected with DV3P12/08 (*n* = 4). Dot color represents scaled mean expression. Scaling was relative to each gene's expression for all cells in each annotation selection, that is, cells associated with each column label in the dot plot. **(B)** Uniform Manifold Approximation and Projection plot showing sub-clusters (profile) of neutrophil populations from the small intestines of DV3P12/08-infected mice (*n* = 4). **(C)** Dot plot depicting cytokine receptor and *Mmp8* gene expression (*n* = 4). **(D)** Dot color represents scaled mean expression. Scaling was relative to the expression of each gene for all cells in each annotation selection, that is, cells associated with each column label in the dot plot (*n* = 4).

anesthetized the mice via an intraperitoneal injection of a mixture of medetomidine, midazolam, and butorphanol prior to viral injection.

### Virus and cell culture

The parental DENV strain DV3P12/08, derived from patients infected with DENV-3 in Thailand (Pambudi et al, 2013), was propagated in C6/36 mosquito cells. Culture supernatants were stored at −80°C until use. C6/36 cells were maintained in L-15 medium containing 10% FCS and 0.3% Bacto Tryptose Phosphate Broth (Becton Dickinson). Vero cells were cultured in Eagle's minimum essential medium (Nacalai Tesque) supplemented with 10% FCS.

### Mouse experiments

IFN-*α*/*β*/*γ*RKO mice, lacking both type I and type II IFN receptors (Phanthanawiboon et al, 2016), were bred and housed in ventilated cages and kept under specific pathogen-free conditions. Male and female mice aged 12 wk were used for this study. Mice were anesthetized via intraperitoneal injections of medetomidine, midazolam, and butorphanol tartrate (final concentrations of 0.3, 4, and 5 mg/kg, respectively) and then intraperitoneally challenged with $2 \times 10^6$ focus-forming units of DV3P12/08. The mice were euthanized using isoflurane at the time of sample collection on day 4 post-infection (p.i.).

### Intestinal cell preparation

The entire small intestine, from the duodenum to the small intestine-cecum junction, was excised on day 4 p.i. Mesenteric fat was removed, and the intestine opened longitudinally and washed in phosphate-buffered saline to remove fecal matter. The cleaned small intestine was quickly frozen and stored in liquid nitrogen. Single-cell isolation was performed following the Fixation & Dissociation Protocol provided by 10X Genomics. The frozen intestine was minced using a scalpel in a Petri dish on ice and transferred into a fixation buffer for incubation at 4°C for 16 h. The fixation buffer was then replaced with ice-cold Tissue Dissociation buffer.

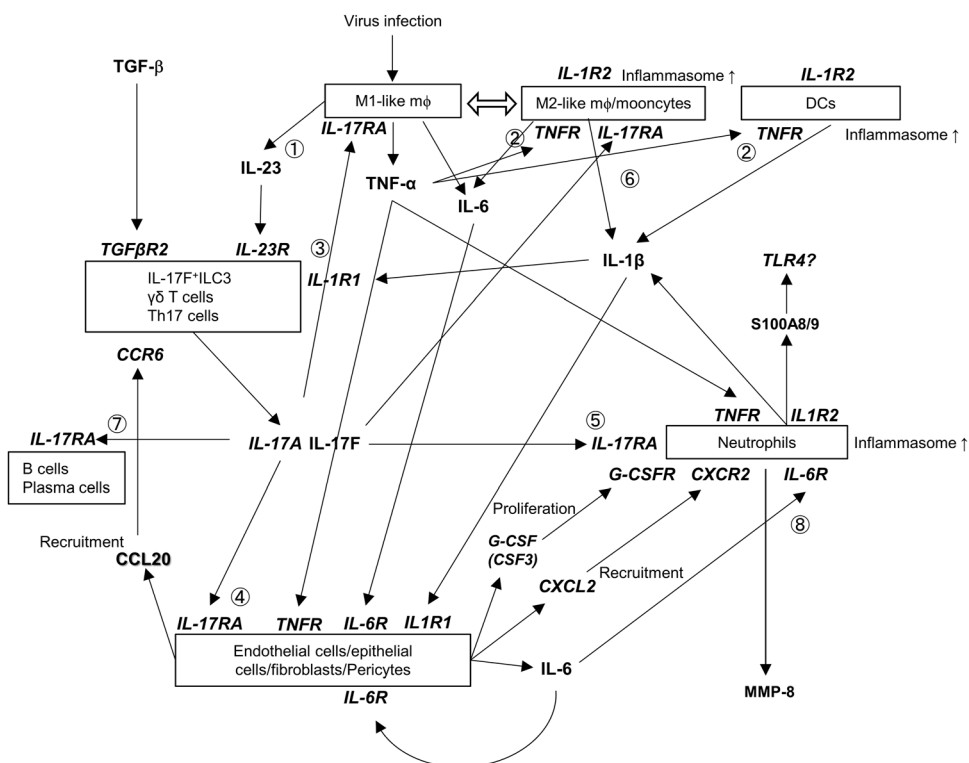

**Figure 9. Schematic depiction of the pathogenic mechanism induced by DENV infection.**
**(1)** Activation of M1-like macrophages during DENV infection leads to the production of TNF-$\alpha$ and IL-23. **(2)** TNF-$\alpha$ stimulates M2-like macrophages and dendritic cells (DCs), resulting in IL-1$\beta$ production. **(3)** IL-1$\beta$ and IL-23 induce IL-17A/F secretion by ILC3s, $\gamma\delta$ T cells, and Th17 cells. **(4)** IL-17A/F, in synergy with other cytokines, such as TNF-$\alpha$ and IL-1$\beta$, activate endothelial cells, epithelial cells, fibroblasts, and pericytes, and induce IL-6 production. **(5)** IL-17A/F promotes the recruitment and activation of neutrophils. **(6)** IL-17A/F stimulates M2-like macrophages. **(7)** IL-17A/F functions in B and plasma cells. **(8)** IL-6 enhances neutrophil activation, whereas neutrophils are further stimulated by CXCL2, G-CSF, and IL-1$\beta$ produced by neutrophils themselves.

After the tissue was transferred into a Dissociation Solution, the intestine was dissociated via a GentleMACS Octo Dissociator (Miltenyi Biotec) according to the following program: incubation for 20 min at 37°C, 50 rpm; spinning for 30 s at 37°C, 2,000 rpm (clockwise); and spinning for 30 s at 37°C, 2,000 rpm (counter-clockwise). The dissociated cells were filtered using a 40 $\mu$m filter and collected via centrifugation at 800 rpm for 2 min. The cells were resuspended in 1 ml chilled Quenching buffer, and then mixed with 0.1 volume of prewarmed Enhancer and 0.275 volume of 50% glycerol. The cells were kept at −30°C until single-cell analysis.

### Single-cell sequencing

RNA from individual fixed cell samples was sequenced using the Fixed RNA Profiling assay (10X Genomics) according to the manufacturer's instructions. Briefly, fixed samples were thawed on ice, mixed with mouse probes, and hybridized with target RNA overnight (18 h) at 42°C. The samples were then washed and counted before being loaded onto the Next GEM Chip Q (10X Genomics) and run on Chromium X (10X Genomics). GEMs were recovered and pre-amplified using PCR. A library was constructed and sequenced on a DNBSEQ-G400RS gene sequencer. The obtained reads were demultiplexed and processed using CellRanger v. 7.1.0 (10X Genomics) with a reference mouse genome (GRCm38 or mm10) and Chromium Mouse Transcriptome Probe Set v1.0.1 mm10. Data preprocessing, clustering, visualization, and differential gene expression were performed using the Python package, Scanpy (version 1.9.1) (Wolf et al, 2018).

### Preprocessing

Cells expressing fewer than 200 genes and genes expressed in less than three cells were excluded. Quality control metrics, such as total counts per cell, number of genes per cell, and proportion of mito-chondrial genes per cell, were calculated. Cells with more than 20% of reads mapped to mitochondrial genes were excluded as low-quality cells. Counts were normalized to a total sum of 10,000 per cell and log-transformed. Highly variable genes were identified and used for further analysis. Finally, undesirable sources of variation (total count and proportion of mitochondrial genes) were regressed using the regress out function, and the data was scaled.

### Clustering

Principal component analysis was performed to reduce dimen-sionality. Batch effect from different samples was corrected and integrated using the Harmony algorithm. We computed the nearest neighbor distance matrix using the scanpy.pp.neighbors function, with the n_neighbors parameter set to 10 and number of principal component analyses set to 30. The Leiden algorithm was used to cluster cells with similar gene expression profiles, and UMAP was used to visualize cell clusters.

### Cell annotation

Cell annotation was automatically performed using the decoupler Python package and canonical mouse markers in the PanglaoDB database. Further correction of the cell annotation of each cluster

was performed manually by checking the canonical markers from literature.

### Enrichment analysis

Gene ontology (GO) enrichment analysis was performed using Gene Set Enrichment Analysis. Differential gene expression was analyzed using the nonparametric Wilcoxon rank-sum test. For the analysis, we included genes with a minimum $\log_2$ fold-change of 1 and $P$-value cutoff of 0.05. We used the GO Biological Process 2023 gene set database. Dot plots were created using the Scanpy.dotplot function.

## Data Availability

The single-cell analysis data of the mouse small intestine reported in this study were deposited in the Sequence Read Archive (SRA) under the accession number: PRJNA1138486.

## Supplementary Information

## Acknowledgements

This work was supported by a grant-in-aid from the Ministry of Education, Culture, Sports, Science, and Technology (MEXT) of Japan (15K148885, 17K08145, 21K05981, and 23K14170) and by grants from the Japan Agency for Medical Research and Development (AMED) (20fk0108404h0001, and 243fa827031j0201), the Mitsubishi Foundation, and the NIPPON Foundation for Social Innovation. We acknowledge Editage (Tokyo, Japan) for their editorial support.

### Author Contributions

M Al Kadi: conceptualization, data curation, software, formal analysis, validation, investigation, visualization, methodology, and writing—original draft, review, and editing.
M Yamashita: data curation, methodology, and writing—review and editing.
M Shimojima: methodology and writing—review and editing.
T Yoshikawa: methodology and writing—review and editing.
H Ebihara: conceptualization and writing—review and editing.
D Okuzaki: conceptualization, resources, software, formal analysis, supervision, funding acquisition, validation, investigation, visualization, methodology, project administration, and writing—review and editing.
T Kurosu: conceptualization, resources, data curation, formal analysis, supervision, funding acquisition, validation, investigation, visualization, methodology, project administration, and writing—original draft, review, and editing.

### Conflict of Interest Statement

The authors declare that they have no conflict of interest.

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
