## [Reviewer comments · Life Science Alliance]

Life Science Alliance

Cytokine Storm and Vascular Leakage in Severe Dengue: Insights from Single-cell RNA Profiling

Mohamad Kadi, Maika Yamashita, Masayuki Shimojima, Tomoki Yoshikawa, Hideki Ebihara, Daisuke Okuzaki, and Takeshi Kurosu

DOI: <https://doi.org/10.26508/lsa.202403008>

Corresponding author(s): Takeshi Kurosu, National Institute of Infectious Diseases and Daisuke Okuzaki, Osaka University

Review Timeline:

Submission Date:	2024-08-21
Editorial Decision:	2024-10-25
Revision Received:	2025-02-14
Editorial Decision:	2025-03-12
Revision Received:	2025-03-17
Accepted:	2025-03-17

Transaction Report:

October 25, 2024

Re: Life Science Alliance manuscript #LSA-2024-03008

Dr. Takeshi Kurosu
National Institute of Infectious Diseases
Department of Virology I
4-7-1 Gakuen
Musashimurayama, Tokyo 208-0011
Japan

Dear Dr. Kurosu,

Thank you for submitting your manuscript entitled "Cytokine Storm and Vascular Leakage in Severe Dengue: Insights from Single-cell RNA Profiling". The manuscript has been evaluated by expert reviewers, whose reports are appended below. Unfortunately, after an assessment of the reviewer feedback, our editorial decision is against publication in Life Science Alliance.

Although your manuscript is intriguing, I feel that the points raised by the reviewers are more substantial than can be addressed in a typical revision period. If you wish to expedite publication of the current data, it may be best to pursue publication at another journal.

Given the interest in the topic, I would be open to re-submission to Life Science Alliance of a significantly revised and extended manuscript that fully addresses the reviewers' concerns and is subject to further peer review. If you would like to resubmit this work to Life Science Alliance, you may submit an appeal directly through our manuscript submission system. Please note that priority and novelty would be reassessed at re-submission.

Regardless of how you choose to proceed, we hope that the comments below will prove constructive as your work progresses.

Thank you for thinking of Life Science Alliance as an appropriate place to publish your work.

Sincerely,

Reviewer #1 (Comments to the Authors (Required)):

In this manuscript, Kadi et al. investigate the pathogenesis of dengue virus-associated cytokine storm. The manuscript follows up on earlier findings identifying TNF- α , IL-6 and MMP-8 as key mediators of cytokine storm in a mouse model of dengue. The authors had identified a role of the small intestine as being important for lethality in DENV-infected mice. In this study, the authors used single-cell RNA sequencing to identify the cell populations in the small intestine that regulate cytokine storm in DENV-infected mice. Overall, the small intestines of infected mice had decreased numbers of neural cells and increased numbers of immune cells (particularly neutrophils) compared to the mock-infected mouse. Pathway analysis revealed an increase in the TNF, NF- κ B and IL-17 signalling pathways, consistent with previous published findings.

The authors then sought to identify which specific cell types contribute to the observed proinflammatory response. They identified T cell populations with the highest IL-17 expression, which correlated with expression of the TGF- β , IL-1 β and IL-23 receptors in these cells, suggest a role for these cytokines in inducing IL-17 expression in the T cells. The authors then identified a role for the CCR6-CCL20 axis in recruiting these T cell populations to the small intestine. Next, the authors investigated IL-6 production in the small intestine and identified endothelial cells, fibroblasts, macrophages pericytes as major producers of IL-6. As drivers of IL-6 expression, the TNF- α receptor was found on all these cell types, while the expression of IL-18 receptor or IL-1 receptor subunits was variable. Thus, the authors suggest that TNF- α is important for the production of IL-6. Macrophages (but not M1-like macrophages), dendritic cells and neutrophils were key producers of IL-1 β , likely also driven by TNF- α signaling. Components of the inflammasome (e.g., NLRP3) was upregulated in macrophages and neutrophils, reflecting a role for the inflammasome in IL-1 β signaling. Finally, MMP-8 was produced by neutrophils and based on receptor expression this could be driven by TNF- α or IL-6.

Overall, this descriptive study identifies cell types expressing important inflammatory mediators during DENV infection in this mouse model, which could open perspectives for future mechanistic studies to investigate DENV pathogenesis and identify therapeutics.

Specific comments

1. Figure 1: Only 1 mock-infected mouse was used compared to 4 DENV-infected mice - increasing the sample size of the mock-infected mice would increase confidence in the findings when comparing the infected mice versus the mock mice.
2. The mice used in this study lack both type I and type II interferon receptors. Can the authors comment on how the lack of an interferon response may affect the physiological relevance of their findings in the context of human infection.

Minor comments

1. Figure 1: It would be helpful to label on the figure panels which are infected mice vs the mock mouse.
2. Line 141 - should be "activate the production of IL-17"

Reviewer #2 (Comments to the Authors (Required)):

The study investigates the pathophysiology of severe dengue, particularly focusing on the cytokine storm and vascular leakage that lead to severe disease outcomes. Using a mouse model, the researchers applied single-cell RNA sequencing to analyze immune responses in the small intestine, identifying key cytokine and effector molecule-producing cells. The research provides new insights into the complex cellular interactions and signaling pathways that drive severe dengue symptoms. This detailed single-cell approach helped elucidate the roles of various immune cells and cytokines in disease progression. The study was performed in an appropriate way and scRNAseq was standard. However major problem in the study is mice model. Mice do not naturally replicate the full spectrum of human dengue disease due to differences in immune system architecture. The mouse immune response, while useful for studying some aspects of pathogenesis, may not fully mimic the cytokine responses or vascular effects seen in humans.

Major Comments:

1. Please discuss the use of the mice model in introduction and in the discussion.
2. In the method section it says male and female mouse used. Total 5 mice were used. It's very difficult to follow. Is four mice infected and one mice control? If yes what is the sex of the mouse in control.
3. Please discuss the limitation of the study for non-human translation.

Reviewer #3 (Comments to the Authors (Required)):

In this study, the investigators have employed a single cell RNA sequencing approach to examine the gene expressions of key cytokines and effector molecule-producing cells, and their receptor expression in the small intestine of a symptomatic DENV3 mouse model using IFN- $\alpha/\beta/\gamma$ -R KO mice. Based on the gene expression profiles observed, the authors claim i) the importance of IL-1 β for the activation of IL-17A-producing cells, (ii) a critical role of IL-17A and TNF- α signalling on T cells, and (iii) infiltrating neutrophil exclusively expressed MMP-8 (Matrix Metalloproteinase 8), a collagen cleaving enzyme which is a major player of inflammatory response.

Building on their previous study that has employed bulk RNA seq, this manuscript solely relies on single cell RNA seq approach to draw conclusions, which is a major limitation of this work. Furthermore, the relevance of the animal model employed is very questionable: IP administration of a high dose of virus. No systemic disease. At day 4 pi. the animals are moribund, which does not mimic the disease kinetic in dengue patients. In such acute model, the clinical manifestations observed are limited to innate immune responses, which is more akin to a sepsis model than a dengue model. In dengue patients, severe clinical manifestations (vascular leakage, shock, etc) occur during the defervescence phase of the disease, after the peak of viremia. There has been increasing evidence of a role of dysfunctional adaptive immune responses in dengue pathogenesis, which this mouse model does not recapitulate. Another example of the lack of clinical relevance of this mouse model is the observation of increased T cell subsets at moribund stage (Fig. 1E&F). Dengue infection is known to cause lymphopenia in human and other mouse models.

-The Discussion is also inadequate. This section mostly repeated the results, with very limited attempt to compare the observations made as part of this work with what has been described in dengue patients. For example, the authors should discuss their speculative hypotheses on the role of pro-inflammatory cytokines in dengue disease severity to the light of the literature, in both dengue patients and other mouse models. For example, in a maternal antibody-mediated ADE mouse model (<https://doi.org/10.1371/journal.pntd.0004536>), treatment with an anti-IL6 blocking mAb was not protective whereas anti-TNF mAb treatment protected the mice.

Other comments:

-Introduction:

1. Page 4, line 42. Inappropriate reference. This should be Bhatt et al. 2013 Nature.

-Results:

1. Figures 2-6. Missing the mock-infected control group for all the figures. Also, I would suggest to present a heat map to include the individual sample including the mock-infected control for fair comparison and clarity.
2. Page 8, line 145. How do you define the naïve CD4 T cells? There is no indication of the annotation.
3. Page 9, lines 175-176. "These results suggest that the CCR6-CCL20 axis mediates the migration of IL-17A/F-producing cells". This is inconclusive as it is not clearly indicated whether these are IL-17A/F-producing cells in figure (3D).
4. Page 10, line 188. Do the authors observe upregulation of the NF- κ B gene in these IL-6 expressing cells?
5. Figure 5E. Sub-cluster A expressed the M2 gene markers more strongly than sub-cluster C. Based on gene expression marker in supplementary Fig 6, sub-cluster C can also be classified as monocytes. Please check these classifications again.
6. Page 12, lines 241-242. Elevated antiviral gene expression does not mean that it is a target for viral infection. It could also mean that sub-cluster D responds more strongly to viral infection in terms of the inflammatory response.
7. Page 12, line 254. Please indicate which figure the authors are referring to.
8. Page 13. Lines 266-267. Please indicate which figure the authors are referring to.
9. Based on the dataset shown in Fig 6A, MMP9 is upregulated in neutrophils from the infected mice. However, the authors claim that in their previous study (Kurosu et al., 2013), they did not find MMP9 expression was induced during infection. The authors should explain why such discrepancy.

Discussion

1. Page 15, line 303. How different are these signals from non-severe dengue? Authors should include a non-severe dengue (using a sub-lethal infectious dose) as a control to explicitly differentiate the pathogenesis of these 2 conditions.
2. Page 17, lines 366-367. "Taken together, these results indicate that macrophages initiate a cytokine storm in the first stage". This is inconclusive since the data are derived from a late stage infection. Analysis at an earlier time point would need to be conducted to support this statement.
3. Page 18, line 382. This is supposed to be supplementary Fig 13.
4. Figure 7. The authors missed the window of the initial event since this is a late stage of infection. Preferably the event should be investigated at an earlier time point where the viremia peaks.

Methodology

1. Page 22, lines 473-474. Were the mice perfused before organ harvesting?
2. Have the authors tried to identify DENV-infected cells using DENV-specific probes?

Reviewer #1 (Comments to the Authors (Required)):

In this manuscript, Kadi et al. investigate the pathogenesis of dengue virus-associated cytokine storm. The manuscript follows up on earlier findings identifying TNF- α , IL-6 and MMP-8 as key mediators of cytokine storm in a mouse model of dengue. The authors had identified a role of the small intestine as being important for lethality in DENV-infected mice. In this study, the authors used single-cell RNA sequencing to identify the cell populations in the small intestine that regulate cytokine storm in DENV-infected mice. Overall, the small intestines of infected mice had decreased numbers of neural cells and increased numbers of immune cells (particularly neutrophils) compared to the mock-infected mouse. Pathway analysis revealed an increase in the TNF, NF- κ B and IL-17 signalling pathways, consistent with previous published findings.

The authors then sought to identify which specific cell types contribute to the observed proinflammatory response. They identified T cell populations with the highest IL-17 expression, which correlated with expression of the TGF- β , IL-1 β and IL-23 receptors in these cells, suggest a role for these cytokines in inducing IL-17 expression in the T cells. The authors then identified a role for the CCR6-CCL20 axis in recruiting these T cell populations to the small intestine. Next, the authors investigated IL-6 production in the small intestine and identified endothelial cells, fibroblasts, macrophages pericytes as major producers of IL-6. As drivers of IL-6 expression, the TNF- α receptor was found on all these cell types, while the expression of IL-18 receptor or IL-1 receptor subunits was variable. Thus, the authors suggest that TNF- α is important for the production of IL-6. Macrophages (but not M1-like macrophages), dendritic cells and neutrophils were key producers of IL-1 β , likely also driven by TNF- α signaling. Components of the inflammasome (e.g., NLRP3) was upregulated in macrophages and neutrophils, reflecting a role for the inflammasome in IL-1 β signaling. Finally, MMP-8 was produced by neutrophils and based on receptor expression this could be driven by TNF- α or IL-6.

Overall, this descriptive study identifies cell types expressing important inflammatory mediators during DENV infection in this mouse model, which could open perspectives for future mechanistic studies to investigate DENV pathogenesis and identify therapeutics.

Specific comments

1. Figure 1: Only 1 mock-infected mouse was used compared to 4 DENV-infected mice - increasing the sample size of the mock-infected mice would increase confidence in the findings when comparing the infected mice versus the mock mice.

>response

We increased the number of mock-infected mice. Although there is not much change from previous observations, we noticed several things, such as mock-infected mice do not have M1-macrophages and there are almost no neutrophils in mock-infected mice. We included these observations and completely rewrote the manuscript.

2. The mice used in this study lack both type I and type II interferon receptors. Can the authors comment on how the lack of an interferon response may affect the physiological relevance of their findings in the context of human infection.

>response

We appreciate your comment. This question has been discussed in the dengue research community and elsewhere. Innate immunity is also important for the establishment of acquired immunity. Although these mice lack major IFN signaling, its effect on acquired immunity is unknown. However, these mice can produce antibodies (Ramadhany et al, Antiviral Research, 2015) and immunization with appropriate antigens protects these mice. These observations suggest that the absence of IFN signaling has little effect on acquired immunity. In the case of human dengue virus infection, patients recover from the due to acquired immunity. These mouse models are not very different in this respect. In the case of innate immunity, we still do not really know how it is involved in disease progression or prevention in patients. Overall, we believe that the mouse model mimics some parts of human disease, but not all. However, it is useful to study mouse models. Even though knockout mouse models can be used. They can be used to study the limitations of human research, for example at the organ level. Both should be studied in a complementary way. In addition, this model is perhaps even more useful as it allows the clearest observation of plasma leakage.

Minor comments

1. Figure 1: It would be helpful to label on the figure panels which are infected mice vs the mock mouse.

>response

Thank you. We have labeled infected and mock-infected in the UMAP analyses. We also described this in the figure legends.

2. Line 141 - should be "activate the production of IL-17"

>response

It meant both IL-17A and IL-17F. Then, we used the word IL-17A/F.

Reviewer #2 (Comments to the Authors (Required)):

The study investigates the pathophysiology of severe dengue, particularly focusing on the cytokine storm and vascular leakage that lead to severe disease outcomes. Using a mouse model, the researchers applied single-cell RNA sequencing to analyze immune responses in the small intestine, identifying key cytokine and effector molecule-producing cells. The research provides new insights into the complex cellular interactions and signaling pathways that drive severe dengue symptoms. This detailed single-cell approach helped elucidate the roles of various immune cells and cytokines in disease progression. The study was performed in an appropriate way and scRNAseq was standard. However major problem in the study is mice model. Mice do not naturally replicate the full spectrum of human dengue disease due to differences in immune system architecture. The mouse immune response, while useful for studying some aspects of pathogenesis, may not fully mimic the cytokine responses or vascular effects seen in humans.

Major Comments:

1. Please discuss the use of the mice model in introduction and in the discussion.

>response

We appreciate your suggestion. We have added the limitation of the mouse study in the discussion.

2. In the method section it says male and female mouse used. Total 5 mice were used. It's very difficult to follow. Is four mice infected and one mice control? If yes what is the sex of the mouse in control.

>response

We have added 3 more mock-infected mice to the analysis. Now, each group, infected or mock-infected has four mice. Two male and 2 female mice for infected mice. Two male and 2 female mice for the mock-infected mice. We haven't seen any difference between male and female in terms of disease progression using this model.

3. Please discuss the limitation of the study for non-human translation.

>response

We have included the limitation of the mouse study in the discussion.

Reviewer #3 (Comments to the Authors (Required)):

In this study, the investigators have employed a single cell RNA sequencing approach to examine the gene expressions of key cytokines and effector molecule-producing cells, and their receptor expression in the small intestine of a symptomatic DENV3 mouse model using IFN- $\alpha/\beta/\gamma$ -R KO mice. Based on the gene expression profiles observed, the authors claim i) the importance of IL-1 β for the activation of IL-17A-producing cells, (ii) a critical role of IL-17A and TNF- α signalling on T cells, and (iii) infiltrating neutrophil exclusively expressed MMP-8 (Matrix Metalloproteinase 8), a collagen cleaving enzyme which is a major player of inflammatory response.

Building on their previous study that has employed bulk RNA seq, this manuscript solely relies on single cell RNA seq approach to draw conclusions, which is a major limitation of this work. Furthermore, the relevance of the animal model employed is very questionable: IP administration of a high dose of virus. No systemic disease. At day 4 pi. the animals are moribund, which does not mimic the disease kinetic in dengue patients. In such acute model, the clinical manifestations observed are limited to innate immune responses, which is more akin to a sepsis model than a dengue model. In dengue patients, severe clinical manifestations (vascular leakage, shock, etc) occur during the defervescence phase of the disease, after the peak of viremia. There has been increasing evidence of a role of dysfunctional adaptive immune responses in dengue pathogenesis, which this mouse model does not recapitulate.

>response

We totally agree that this study has the limitation. Mouse study may not reflect human disease. However, we believe that some parts of the pathogenic mechanism of mouse models reflect those of human infection. This mouse model shows clear vascular leakage, which is the most unique and important symptom of severe dengue. In addition, dengue virus infects macrophages (Kurosu et al, PLoS Neglected Tropical Diseases, 2023), which is consistent with human cases. This mouse model can produce protective anti-dengue virus antibodies by at least day 6 post-infection (Ramadhany et al, Antiviral Research, 2015). In this model, the mice died on day 4-5 post-infection. This gives the

impression that this is too early. However, the peak of virus production is earlier than day 4. In case of human dengue, we do not exactly know whether the virus gets complete clearance from the body at the same time although peripheral blood shows a decrease in virus titer on the day of shock. Several dengue researchers have suggested that even when virus is cleared from the peripheral blood, some cells in some organs continue to produce viruses causing vascular leakage. Regarding sepsis, several groups believe that the pathogenesis of sepsis is partly shared with the pathogenesis of dengue (Aguilar-Briseno et al, Current Opinion in Virology, 2020). We also believe that some parts of the sepsis mechanism are similar because similar host molecules are involved in disease progression. Therefore, if our model is similar to sepsis, it means that this model reflects the disease of severe dengue.

As a reviewer mentioned, dengue is a systemic disease caused by a systemic infection. This means that the infection affects the whole body, not just one organ or part of body. In this respect, we think that this model shows the systemic infection. This often happens because circulating blood cells are infected. In our model, Iba1⁺ macrophages were infected (Kurosu et al, PLoS Neglected Tropical Diseases, 2023). Mice in this model die from a cytokine storm because anti-TNF- α treatment protects 100% of mice. Vascular leakage was observed not only in the small intestine, but also in the liver, etc (Phanthanawiboon et. al., PLoS One 2016). It is the result of several systemic processes.

Another example of the lack of clinical relevance of this mouse model is the observation of increased T cell subsets at moribund stage (Fig. 1E&F). Dengue infection is known to cause lymphopenia in human and other mouse models.

>response

We understand the reviewer's point. Regarding lymphopenia, it is true that patients often have mild lymphopenia. This seem discrepant with our result. However, this may be an important feature to understand our observation. We need to consider this separately from our current result because one is a peripheral event in the blood and the other is a local event in the organs. Lymphopenia may be the result of the movement of lymphocytes from peripheral blood to local organs although nobody knows that. In humans, expansion of skin-homing lymphocytes was observed one day before defervescence (Arora et al, iScience, 2022). In addition, IL-17 producing cells such as Th17 cells, $\gamma\delta$ T cells, and ILCs are a small population, especially in peripheral blood. ILCs probably cannot be originally detected in the peripheral blood in humans. Increase or decrease of their numbers has little effect on the counts of lymphocytes in peripheral

blood.

-The Discussion is also inadequate. This section mostly repeated the results, with very limited attempt to compare the observations made as part of this work with what has been described in dengue patients. For example, the authors should discuss their speculative hypotheses on the role of pro-inflammatory cytokines in dengue disease severity to the light of the literature, in both dengue patients and other mouse models. For example, in a maternal antibody-mediated ADE mouse model (<https://doi.org/10.1371/journal.pntd.0004536>), treatment with an anti-IL6 blocking mAb was not protective whereas anti-TNF mAb treatment protected the mice.

>response

We appreciate the reviewer's comments. We totally rewrote the discussion to avoid repeating the results. Regarding blocking to TNF or IL-6, by using the same mouse model in this study, we demonstrated the reason why the blocking IL-6 does not efficiently protect mice from lethal infection with dengue virus (Kurosu et al, PLoS Neglected Tropical Diseases, 2023). This is because the number (concentration) of IL-6 molecules is too high compared to that of TNF- α . By monitoring the downstream molecule, serum albumin A 3 (SAA3), of IL-6, we were able to show this. As we have already reported this (Kurosu et al, PLoS Neglected Tropical Diseases, 2023), we did not write about IL-6.

Other comments:

-Introduction:

1. Page 4, line 42. Inappropriate reference. This should be Bhatt et al. 2013 Nature.

>response

Thank you for your correction. We replaced it with this reference.

-Results:

1. Figures 2-6. Missing the mock-infected control group for all the figures. Also, I would suggest to present a heat map to include the individual sample including the mock-infected control for fair comparison and clarity.

>response

We added 3 more mock-infected control mice, so a total of 4 mock-infected mice. And show them separately. Necessary heatmaps are shown in supplementary figures.

2. Page 8, line 145. How do you define the naïve CD4 T cells? There is no indication of the annotation.

>response

We analyzed T cells and ILCs again. Annotation is in Supplementary Figure S2.

3. Page 9, lines 175-176. "These results suggest that the CCR6-CCL20 axis mediates the migration of IL-17A/F-producing cells". This is inconclusive as it is not clearly indicated whether these are IL-17A/F-producing cells in figure (3D).

>response

We appreciate your question. It is an interesting point. If CCR6 expression is detected in the cells that express IL-17A/F, it makes sense. However, even if only a small percentage of CCR6-expressing cells express IL-17A/F, we cannot deny the importance of the CCR6-CCL20 axis. Recruitment of cells through CCR6-CCL20 axis and IL-17A/F production do not have to be linked because they are different events through different biological processes. That's why we just wanted to show that the CCR6-CCL20 axis is used for the recruitment of IL-17-producing "candidate" cells. This figure is more like "We have confirmed that the CCR6-CCL20 axis works in severe disease". We thought this was useful information because there is no report of this kind of report using a real severe infection animal model at the single cell level. If it is still not enough, we will delete this result.

4. Page 10, line 188. Do the authors observe upregulation of the NF- κ B gene in these IL-6 expressing cells?

>response

Thank you for your question. Technically, it is difficult to detect both of them at single cell level because the activation of NF- κ B needs to be detected by another way, different from transcriptional level. We cannot directly detect activation of NF- κ B by analyze mRNA expression. What we can do is to predict its activation judging from increased expression of downstream genes. IL-6 is one of the known genes activated by NF- κ B signaling. We showed that the NF- κ B is activated by infection at the tissue level in the small intestine (Kurosu et al, PLoS Neglected Tropical Diseases, 2023). In a previous report, we demonstrated that the phosphorylated p65 subunit of NF- κ B was translocated into the nucleus of stroma-like cells, including endothelial cells, immune cells, and intestinal epithelial cells by infection in this model. This indicates the activation of NF- κ B.

5. Figure 5E. Sub-cluster A expressed the M2 gene markers more strongly than sub-cluster C. Based on gene expression marker in supplementary Fig 6, sub-cluster C can also be classified as monocytes. Please check these classifications again.

>response

Thank you for your suggestion. We re-analyzed the sub-cluster again by combining data from four mock-infected mice. In the new version of sub-cluster, macrophages were classified into three groups, M1-like macrophages, M2-like macrophages, and monocytes.

6. Page 12, lines 241-242. Elevated antiviral gene expression does not mean that it is a target for viral infection. It could also mean that sub-cluster D responds more strongly to viral infection in terms of the inflammatory response.

>response

We agree. That's why we used the word, a "potential" target.

7. Page 12, line 254. Please indicate which figure the authors are referring to.

>response

We have added the figure number.

8. Page 13. Lines 266-267. Please indicate which figure the authors are referring to.

>response

We have added the figure number.

9. Based on the dataset shown in Fig 6A, MMP9 is upregulated in neutrophils from the infected mice. However, the authors claim that in their previous study (Kurosu et al., 2013), they did not find MMP9 expression was induced during infection. The authors should explain why such discrepancy.

>response

Thank you for your suggestion. There is no discrepancy. Neutrophils from mock-infected mice express MMP-9 at basal levels (Kurosu et al, PLoS Neglected Tropical Diseases, 2023). Unfortunately, we still show the same result because there was only one neutrophil was detected in mock-infected mice. Although a fixed single-cell RNA profiling assay has an advantage, it also has a disadvantage. This is the limitation of the number of cell that can be analyzed. This method cannot handle many cells, the maximum number being around 10,000 cells. As neutrophils are not a major population in the small intestine of mock-infected mice, neutrophil could not able to be

detected in mock-infected mice.

Discussion

1. Page 15, line 303. How different are these signals from non-severe dengue?

Authors should include a non-severe dengue (using a sub-lethal infectious dose) as a control to explicitly differentiate the pathogenesis of these 2 conditions.

>response

Thank you for your suggestion. It may be useful to compare these groups. This time our main aim is to characterize each cell (or cluster) by examining cytokine, chemokine, and their receptor expressions. We did not think that we could obtain clear data by studying a mild disease model. In a previous report using a similar type of mouse model, a slightly milder model showed slightly lower IL-6 mRNA activation compared to the most severe disease (Frontiers in Microbiology), and avirulent virus showed lower virus titers in some organs. These observations led us to think the inclusion of mild group would be complicated. In order to simplify the result and to obtain a clear expression of the host gene, in this study we focused on the comparison between infected and mock-infected mouse groups.

2. Page 17, lines 366-367. "Taken together, these results indicate that macrophages initiate a cytokine storm in the first stage". This is inconclusive since the data are derived from a late stage infection. Analysis at an earlier time point would need to be conducted to support this statement.

>response

We agree. However, based on the expression of cytokines and their receptors, we can guess the order of activation. Of course, further analysis is needed, but an indication or direction from the current observation is important for further research. That's why we used the word,"indicate."

3. Page 18, line 382. This is supposed to be supplementary Fig 13.

>response

Thank you for pointing this out. We have corrected it.

4. Figure 7. The authors missed the window of the initial event since this is a late stage of infection. Preferably the event should be investigated at an earlier time point where the viremia peaks.

>response

We will do it next time. As mentioned above, this time, we focused on studying the expression of cytokines, chemokines, and their receptors to identify the responsible cells in order to understand the vascular mechanism.

Methodology

1. Page 22, lines 473-474. Were the mice perfused before organ harvesting?

>response

No, we didn't. The point of this experiment is to recover the small intestine as quickly as possible. Instead, we washed it with ice-cold PBS, and quickly froze it in liquid nitrogen.

2. Have the authors tried to identify DENV-infected cells using DENV-specific probes?

>response

No, not this time. We will try it for further studies. However, as mentioned above, we know that macrophages are the exclusive target of infection in this model (Kurosu et al, PLoS Neglected Tropical Diseases, 2023).

March 12, 2025

RE: Life Science Alliance Manuscript #LSA-2024-03008R-A

Dr. Takeshi Kurosu
National Institute of Infectious Diseases
Department of Virology I
4-7-1 Gakuen
Musashimurayama, Tokyo 208-0011
Japan

Dear Dr. Kurosu,

Thank you for submitting your revised manuscript entitled "Cytokine Storm and Vascular Leakage in Severe Dengue: Insights from Single-cell RNA Profiling". We would be happy to publish your paper in Life Science Alliance pending final revisions necessary to meet our formatting guidelines.

- please respond to Reviewer 3's remaining comments
- please be sure that the authorship listing and order is correct
- please upload all figure files as individual ones, including the supplementary figure files; all figure legends should only appear in the main manuscript file
- please add the X and Bluesky handles of your host institute/organization as well as your own or/and one of the authors in our system
- please add Author Contributions to our system as well
- please add your main, supplementary figure, and table legends to the main manuscript text after the references section
- there is a callout for table S3, and it hasn't been uploaded to the system...please correct

A. FINAL FILES:

B. MANUSCRIPT ORGANIZATION AND FORMATTING:

Sincerely,

Reviewer #1 (Comments to the Authors (Required)):

Previous concerns have been addressed through additional experiments (e.g. inclusion of more mock-infected mice) and discussion (e.g. limitations of murine model). Although the findings are descriptive in nature, they could open perspectives for future mechanistic studies to investigate DENV pathogenesis and thus are likely to be of interest to readers.

Reviewer #3 (Comments to the Authors (Required)):

The authors have re-analysed the single-cell RNAseq data and included more uninfected controls, which makes the claims and findings more convincing. However, the main limitation of this paper remains which is that they entirely draw their conclusions based on single cell RNAseq, with no experimental validation. The relatively small number of cells analysed by single cell RNA seq (10,000) also makes the findings less convincing.

In their rebuttal to justify the relevance of the mouse model employed, the authors argue that this mouse model reflects the systemic severe dengue. This Reviewer then questions why did they look at the immune responses in the intestines and not in the blood?

Reviewer #3 (Comments to the Authors (Required)):

The authors have re-analysed the single-cell RNAseq data and included more uninfected controls, which makes the claims and findings more convincing. However, the main limitation of this paper remains which is that they entirely draw their conclusions based on single cell RNAseq, with no experimental validation. The relatively small number of cells analysed by single cell RNA seq (10,000) also makes the findings less convincing.

In their rebuttal to justify the relevance of the mouse model employed, the authors argue that this mouse model reflects the systemic severe dengue. This Reviewer then questions why did they look at the immune responses in the intestines and not in the blood?

>response

Thank you for your question. We appreciate your perspective and would like to explain why we focused on the small intestine.

First, we examined organs that exhibit clear vascular leakage because our goal was to understand the pathogenic mechanisms leading to vascular leakage, rather than merely observing immune cell activation. This is why we did not analyze the blood. Previous studies have shown that the liver and intestine experience the most severe vascular leakage (Phanthanawiboon et al., 2016). We specifically chose the small intestine because the damage observed there appears to be more critical for survival (Kurosu et al., 2023). In this model, blocking TNF- α signaling protects mice, and this treatment has a more pronounced ameliorative effect in the small intestine. Additionally, pro-inflammatory cytokine levels are significantly higher in the small intestine compared to the liver. Moreover, IL-17 plays a crucial role in disease progression in this model, and its production was observed in the small intestine but not in the liver.

Second, it is well known that patients with severe dengue frequently experience gastrointestinal symptoms (Chang et al., 2017; Sam et al., 2013), although this aspect has been somewhat underestimated. These symptoms are often thought to be merely a consequence of vascular leakage. However, it is important to recognize that the intestinal tract is the largest organ where immune cells reside (Mowat and Agace, 2014; Chassaing et al., 2014). We believe it plays a significant role in disease progression, particularly in cytokine production, and we wanted to highlight this aspect.

For these reasons, we focused on the small intestine in our study. As mentioned above, we agree that most clinical studies have analyzed immune responses using peripheral blood. We will consider investigating blood cells in future studies.

Regarding the number of cells, we understand Reviewer #3's concerns, but we would like to clarify that this method is relatively new. What we performed was not traditional single-cell RNA sequencing; rather, we conducted a Fixed RNA Profiling assay. Although the number of cells analyzed in this method is generally lower compared to single-cell RNA sequencing, in our study, we analyzed more than 40,000 cells. Additionally, we included four infected and four uninfected animals, ensuring the reliability of our results. Furthermore, this method offers several advantages. It allows for the analysis of fragile cells by enabling immediate fixation in liquid nitrogen. In our previous study, we were able to detect only $\gamma\delta$ T cells as IL-17 producers using flow cytometry. However, in the current study, we successfully identified two additional IL-17-producing cell types. We believe that these cells may have been lost during the cell preparation steps in our prior experiments. In fact, we observed significant cell loss during certain stages of the preparation process. We hope that future advancements in this technology will allow for an even greater number of cells to be analyzed, enabling more comprehensive investigations

March 17, 2025

RE: Life Science Alliance Manuscript #LSA-2024-03008RR

Dr. Takeshi Kurosu
National Institute of Infectious Diseases
Department of Virology I
4-7-1 Gakuen
Musashimurayama, Tokyo 208-0011
Japan

Dear Dr. Kurosu,

Thank you for submitting your Resource entitled "Cytokine Storm and Vascular Leakage in Severe Dengue: Insights from Single-cell RNA Profiling". It is a pleasure to let you know that your manuscript is now accepted for publication in Life Science Alliance. Congratulations on this interesting work.

DISTRIBUTION OF MATERIALS:

Again, congratulations on a very nice paper. I hope you found the review process to be constructive and are pleased with how the manuscript was handled editorially. We look forward to future exciting submissions from your lab.

Sincerely,
